# Learning to Bet for Horizon-Aware Anytime-Valid Testing

Ege Onur Taga [1]  Samet Oymak [1]  Shubhanshu Shekhar [1]

## Abstract

We develop *horizon-aware anytime-valid* tests and confidence sequences for bounded means under a strict deadline $N$. Using the betting/e-process framework, we cast horizon-aware betting as a finite-horizon optimal control problem with state space $(t, \log W_t)$, where $t$ is the time and $W_t$ is the test martingale value. We first show that in certain interior regions of the state space, policies that deviate significantly from Kelly betting are provably suboptimal, while Kelly betting reaches the threshold with high probability. We then identify sufficient conditions showing that outside this region, more aggressive betting than Kelly can be better if the bettor is behind schedule, and less aggressive can be better if the bettor is ahead. Taken together these results suggest a simple phase diagram in the $(t, \log W_t)$ plane, delineating regions where Kelly, fractional Kelly, and aggressive betting may be preferable. Guided by this phase diagram, we introduce a Deep Reinforcement Learning approach based on a *universal* Deep Q-Network (DQN) agent that learns a single policy from synthetic experience and maps simple statistics of past observations to bets across horizons and null values. In limited-horizon experiments, the learned DQN policy yields state-of-the-art results.

## 1. Introduction

Consider a stream of i.i.d. observations $\{X_n : n \geq 1\}$ drawn from an unknown distribution $P_X$, supported on $\mathcal{X} = [0, 1]$ with mean $\mu_X$. A fundamental task in statistics is testing the null

$$H_0 : \mu_X = m, \quad \text{versus} \quad H_1 : \mu_X \neq m,$$

[1]Department of Electrical and Computer Engineering, University of Michigan, Ann Arbor, USA. Correspondence to: Ege Onur Taga <egetaga@umich.edu>.

*Proceedings of the 43rd International Conference on Machine Learning*, Seoul, South Korea. PMLR 306, 2026. Copyright 2026 by the author(s).

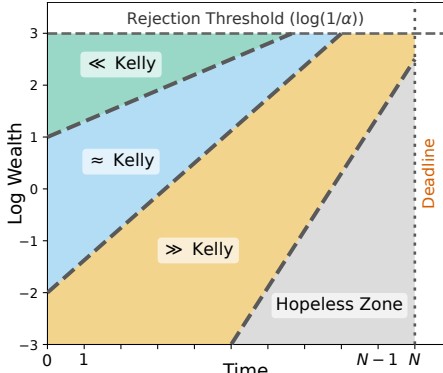

*Figure 1.* A representative phase diagram in the $(t, \log W_t)$ plane, illustrating the qualitative partition of the plane into regimes where Kelly, aggressive, and conservative betting may be preferred under a finite horizon $N$ and significance level $\alpha$. Formal results implying such a partition are presented in Section 3.

for a known value $m \in [0, 1]$. Formally, this involves designing a *level-$\alpha$ test of power one* (Darling & Robbins, 1968), which are stopping times $\tau_m$ satisfying

$$\mathbb{P}_{H_0}(\tau_m < \infty) \leq \alpha, \quad \text{and} \quad \mathbb{P}_{H_1}(\tau < \infty) = 1.$$

In words, $\tau_m$ denotes the random number of observations after which the analyst stops and rejects the null. A related problem is that of constructing confidence sequences for the mean $\mu_X$, which are time-uniform generalizations of confidence intervals, and consist of a sequence of subsets $\{C_n : n \geq 0\}$, with $C_n$ being $\mathcal{F}_n = \sigma(X_1, \ldots, X_n)$-measurable, and satisfying the uniform coverage guarantee:

$$\mathbb{P}(\forall n \geq 0 : \mu_X \in C_n) \geq 1 - \alpha.$$

While these problems have a long history going back to the pioneering works by Robbins and coauthors in the 1960s, there has been a recent surge of interest and activity in this area, led by Shafer & Vovk (2019), Grünwald et al. (2024), Waudby-Smith & Ramdas (2024), and Orabona & Jun (2023). A comprehensive recent survey on this topic can be found in (Ramdas et al., 2023).

In many problems, the analyst may want to look at the data repeatedly and possibly stop early, but the experiment is also constrained by a hard resource limit modeled by a maximum sample size $N < \infty$. This includes applications

such as online A/B tests, adaptive experimentation, and time or resource constrained scientific studies. This leads to a tension between the two classical formulations. A fixed-sample size test designed for the time $N$ can lead to inflated false alarms under continuous monitoring, while a the fully time-uniform procedures protect against indefinite continuous monitoring at the cost of extra conservatism, that may be unnecessary given the hard deadline $N$.

This motivates the design of *horizon-aware* or *deadline-aware* tests and confidence sequences that retain their validity under continuous monitoring up to a prespecified deadline $N$. Concretely we seek to design stopping rules $\tau \leq N$ and confidence sets $\{C_n : 1 \leq n \leq N\}$ such that

$$\sup_{P:\, \mathbb{E}_P[X]=m} \mathbb{E}_P(\tau \leq N) \leq \alpha,$$
$$\inf_{P:\, \mathbb{E}_P[X]=\mu_X} \mathbb{P}_P(\forall n \leq N : \mu_X \in C_n) \geq 1 - \alpha.$$

In addition to the above validity guarantees, we also want to ensure that the inference within the $N$ observations is as sharp as possible. Informally, this demand induces a certain amount of nonstationarity in the design of these procedures that is absent in the usual anytime-valid inference.

## 2. Background

The betting approach to constructing sequential tests and confidence sets is based on the principle of testing by betting, popularized by Shafer (2021) and Shafer & Vovk (2019). This principle states that the evidence against a null hypothesis can be precisely quantified by the gain in wealth achievable by a (fictitious) bettor betting repeatedly on the observations under odds that are fair if the null were true. Thus, if the null is indeed true, then it is mathematically unlikely for the bettor to increase his initial wealth by large multiples irrespective of the betting strategies employed. On the other hand, if the null is not true, then good betting strategies may exploit this and lead to exponentially growing wealth. Thus, in this framework, the problem of statistical inference can be transformed into that of repeatedly betting on the stream of outcomes.

Waudby-Smith & Ramdas (2024) operationalized this principle for the task of constructing power-one tests and confidence sequences for the mean of bounded random variables. In particular, for the null $H_0 : \mu_X = m$, Waudby-Smith & Ramdas (2024) constructed the following process:

$$W_n(m) = \begin{cases} 1, & n = 0, \\ W_{n-1}(m)\big(1 + \lambda_n(m)(X_n - m)\big), & n \geq 1. \end{cases}$$

where $\{\lambda_n(m) : n \geq 1\}$ is a predictable sequence of $[-1/(1-m), 1/m]$ valued "bets" (by predictable, we mean that each $\lambda_n(m)$ is $\sigma(X_1, \ldots, X_{n-1})$-measurable). This

process $\{W_n(m) : n \geq 0\}$ can be interpreted as the evolution of the wealth of a fictitious bettor, who starts with one dollar, and bets repeatedly on the next outcome by selecting the parameter $\lambda_n(m)$. If the null is true, and $\mu_X$ is $m$, then the process $\{W_n(m) : n \geq 0\}$ is a so-called *test martingale* (this is, a nonnegative (super-) martingale with an initial value of 1). Such processes satisfy a time-uniform generalization of Markov's inequality, called Ville's inequality (recalled in Fact A.1 in Appendix A), that says

$$\mathbb{P}_{H_0}(\exists\, n \geq 0 : W_n(m) \geq 1/\alpha) \leq \alpha, \ \forall\, \alpha \in (0, 1].$$

This immediately suggests the following stopping time

$$\tau_m := \inf\{n \geq 1 : W_n(m) \geq 1/\alpha\},$$

satisfying the required level-$\alpha$ property $\mathbb{P}_{H_0}(\tau_m < \infty) \leq \alpha$. Using the standard duality between power-one tests and confidence sequences, such stopping times can be used to construct confidence sequences $\{C_n : n \geq 1\}$ with

$$C_n = \{m \in [0, 1] : \tau_m > n\}.$$

The betting strategy is crucial to the design of tests that perform well under the alternative, and confidence sequences whose width decays quickly. Waudby-Smith & Ramdas (2024) proposed and analyzed several betting strategies, and conducted a thorough empirical study. Orabona & Jun (2023) employed the regret of the universal portfolio algorithm of Cover (1991); Cover & Ordentlich (1996) to construct a computationally feasible betting confidence sequence with strong theoretical and empirical performance. Follow up works building upon the betting approach for bounded random variables include Ryu & Bhatt (2024); Shekhar & Ramdas (2023); Chen & Wang (2025); Clerico (2025).

All these results work under the assumption of an infinite stream of observations, which is natural in view of the definitions of power-one tests and confidence sequences. However, in practical applications, there is often a hard upper limit on the number of observations that can be collected, due to fundamental limitations of time and resources. Placing such hard limits necessitates a rethink of the betting strategies for the construction of tests and confidence sets.

A complementary perspective is presented by Koning & Van Meer (2026), who show that anytime-validity can be induced from a terminal procedure fixed sample size test through a Doob martingale construction. As a result they show that in settings where such constructions are tractable, one can make a classical fixed sample size test valid under continuous monitoring without losing power. Our focus is different from that of Koning & Van Meer (2026). We restrict our attention to a class of multiplicative betting e-processes, and ask how the betting policy should be modified in the presence of a hard deadline $N$.

To the best of our knowledge, the only recent work to consider the task of constructing horizon-aware betting-based confidence intervals and tests is (Voráček & Orabona, 2025). In short, Voráček & Orabona (2025) observed that in the finite horizon regime, the bets at time $t$ must take into account (i) the (log-) distance from the threshold $\log(1/\alpha) - \log W_t$, and (ii) the remaining time $N - t$. Building upon this insight the authors introduce Sequential Target-Recalculating (STaR) betting strategies and show that these can yield strict improvements over betting based Hoeffding and Bernstein-style tests and confidence sets. They further propose a STaR-Bets algorithm, obtained by applying the STaR principle to a variance-adaptive empirical Bernstein type test ("Bets") which is similar in construction to the predictable plug-in empirical-Bernstein betting scheme (PrPlEB) of Waudby-Smith & Ramdas (2024). They prove that the underlying Bets procedure yields a CS whose width at $N$ is within $(1 + o(1))$ factor of the optimal fixed sample-size CI. Our approach, as described in Section 3 differs fundamentally from that of Voráček & Orabona (2025). In particular, we formulate the task of horizon-aware betting as a finite horizon optimal control problem aimed at maximizing the probability of rejection within the horizon $N$ while maintaining anytime-validity. While explicitly identifying the optimal finite horizon policy for this dynamic program (DP) is not tractable in general, we obtain some insights about the optimal policy by considering some specific conditions (discrete distributions, either ahead of or far behind the schedule, etc.). Together these exploratory results allow us to construct a two-dimensional "phase-diagram" identifying regimes in the $(t, \log W_t)$ plane where the optimal strategy is close to Kelly, more aggressive than Kelly, and more conservative than Kelly. Based on these insights we develop a class of computationally efficient heuristics (hedging over $\epsilon$-greedy strategies) and a universal deep-Q-learning (DQN) based betting strategy. Through empirical results we illustrate how these learned betting strategies can lead to improvements in rejection probability and CS widths.

## 3. Horizon-Aware Testing by Betting

We now formally describe the problem of designing horizon-aware power-one tests for the mean of bounded random variables. Let $\{X_n : n \geq 1\}$ denote a sequence of i.i.d. $[0, 1]$-valued random variables, drawn according to a distribution $P_X$ with an unknown mean $\mu_X \in [0, 1]$. Throughout this section, we will assume that $\{\mathcal{F}_n : n \geq 0\}$ represents a filtration on some underlying probability space, such that $\sigma(X_1, \ldots, X_n) \subset \mathcal{F}_n$ for all $n \geq 1$. For a known $m \in [0, 1]$, we are interested in constructing a power-one test $\tau_m$ for the null $H_{0,m} : \mu_X = m$, against the alternative $H_{1,m} : \mu_X \neq m$. Formally, we want to define a $\mathbb{N} \cup \{\infty\}$-valued random variable $\tau_m$ (here, $\mathbb{N} = \{0, 1, \ldots\}$), such

that for a prespecified $\alpha \in (0, 1]$, we have

$$\{\tau_m \leq n\} \in \mathcal{F}_n \quad \text{for all } n \geq 1,$$

$$\mathbb{P}_{H_{0,m}}(\tau_m < \infty) \leq \alpha, \quad \text{and} \quad \mathbb{P}_{H_{1,m}}(\tau_m < \infty) = 1.$$

This is the classical definition of a level-$\alpha$ power-one test. However, in this paper, we assume that we also have a strict prespecified deadline $N \in \mathbb{N}$; that is, we can not collect more than $N$ observations. If $\tau_m = \inf\{n \geq 1 : W_n(m) \geq 1/\alpha\}$ is defined as the first $1/\alpha$ crossing of a test martingale (or more generally an e-process), then we can restate our objective as

$$\begin{aligned}
\text{maximize} \quad & \mathbb{E}_{H_{1,m}}[u(\tau_m)\mathbf{1}_{\tau_m \leq N}] \\
\text{subject to} \quad & \mathbb{E}_{H_{0,m}}[W_\eta(m)] \leq 1,
\end{aligned} \quad (1)$$

where $u$ is any nonnegative and nonincreasing "utility function", and $\eta$ denotes any $[N] := \{1, 2, \ldots, N\}$-valued stopping time satisfying $\{\eta \leq n\} \in \mathcal{F}_n$ for all $n \in [N]$.

Working in the betting framework, we can construct level-$\alpha$ stopping times as

$$\tau_m = \inf\{n \geq 1 : W_n \geq 1/\alpha\}, \quad \text{with } W_0 = 1,$$

$$W_n = W_{n-1}\Big(1 + \lambda_n(m)\,(X_n - m)\Big).$$

where each $\lambda_n(m)$ is $\mathcal{F}_{n-1}$-measurable lying in the range $\Lambda_m = [-1/(1 - m), 1/m]$. In other words, the sequence $\{\lambda_n(m) : n \geq 1\}$ is a predictable sequence with respect to the filtration $\{\mathcal{F}_n : n \geq 0\}$. Since the stopping time $\tau_m$ associated with any sequence of predictable bets lying in $\Lambda_m$ is level-$\alpha$ by construction due to Ville's inequality (see Fact A.1 in Appendix A), we can restate the objective of (1) with $u(t) = 1$ for simplicity as follows:

$$\text{maximize} \quad \mathbb{P}\left(\max_{1 \leq n \leq N} W_n \geq \frac{1}{\alpha}\right),$$

$$\text{s.t.} \quad \begin{cases}
W_0 = 1, \quad W_n = W_{n-1}\left(1 + \lambda_n(m)(X_n - m)\right), \\
\lambda_n \text{ is } \mathcal{F}_{n-1}\text{-measurable for all } n \geq 1, \\
\lambda_n \in \Lambda_m = \left[\dfrac{-1}{1-m}, \dfrac{1}{m}\right] \forall n \geq 1.
\end{cases}$$

For every $P_X$ with $\mu_X \neq m$, the problem described above will have a sequence of optimal bets $\{\lambda_n^*(m) : 1 \leq n \leq N\}$. To formalize this, fix such a $P_X$, and with $Y_t = \log W_t$, and $h_m(\lambda, x) = \log(1 + \lambda(x - m))$, define

$$V_t(y) = \sup_{\{\lambda_i\}_{i=t+1}^N} \mathbb{P}\left(\max_{j \in \{t+1, \ldots, N\}} Y_j \geq b \,\Big|\, Y_t = y\right),$$

$$V_N(y) = \mathbf{1}\{y \geq b\}, \quad b := \log(1/\alpha),$$

where we use the notation $\mathbf{1}\{S\}$ to denote the 0-1-valued indicator of the statement $S$. This suggests the following

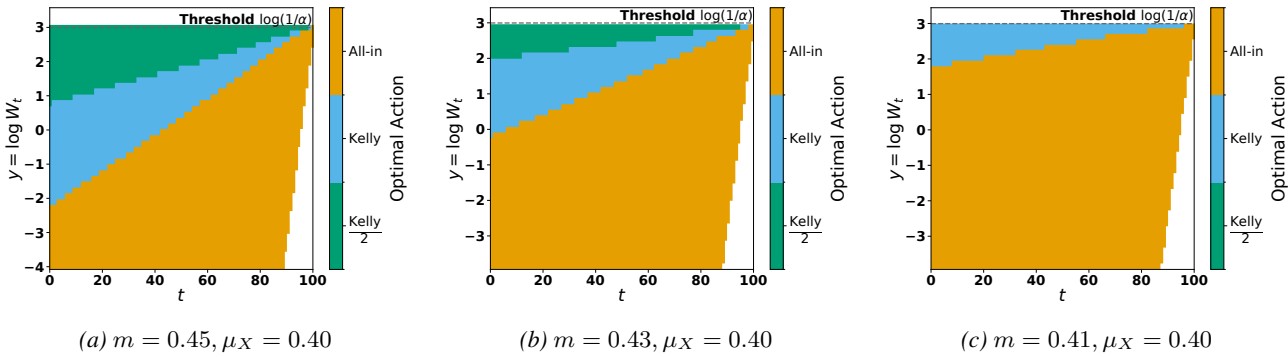

*Figure 2.* The phase diagram of optimal actions in $(t, \log W_t)$ plane demonstrating the change of optimal policies with respect to problem difficulty. $X_i$ drawn from a two-component Beta mixture: $X_i \sim \text{Beta}(2.4, 3.6)$ with probability $1/2$ and $X_i \sim \text{Beta}(4.8, 7.2)$ with probability $1/2$, which has mean $\mu_X = 0.40$ and varying variance across realizations. Experimental details are in Section 4.1.

Bellman recursion:

$$V_t(y) = \max_{\lambda \in \Lambda_m} \mathbb{E}_{X \sim P_X} \Big[ \mathbf{1}\{y + h_m(\lambda, X) \geq \log(1/\alpha)\}$$
$$+ \mathbf{1}\{y + h_m(\lambda, X) < \log(1/\alpha)\} V_{t+1}\big(y + h_m(\lambda, X)\big) \Big]. \tag{2}$$

Instead of attempting to identify the optimal policy exactly, we present some results that provide partial characterization of the optimal policy under certain conditions. Together these results allow us to create an informal "phase diagram" of the behavior of the optimal betting policy that we alluded to in Figure 1.

To state our first result, we need some additional notation. Throughout we will work with $\Lambda$ denoting a compact subset of $\Lambda_m$ that contains the Kelly bet $\lambda_m^{\text{Kelly}}$. For any $\lambda \in \Lambda$, let

$$L(\lambda) = \mathbb{E}[h_m(\lambda, X)], \quad L_{\max} = L(\lambda_m^{\text{Kelly}}), \quad \text{and}$$
$$B := \sup_{\lambda \in \Lambda, \, x \in [0,1]} \big| h_m(\lambda, x) \big| < \infty.$$

**Theorem 3.1.** *Consider any time $t$, and log-wealth $y = \log W_t$. Let $b = \log(1/\alpha)$ denote the threshold, and $T = N - t$ the remaining time. Fix a $\delta > 0$, and assume that there exists an $\epsilon \equiv \epsilon(\delta) > 0$, such that $|\lambda - \lambda_m^{\text{Kelly}}| \geq \delta$ implies $L(\lambda) \leq L_{\max} - \epsilon$. Let $\Delta$ denote the term $TL_{\max} - (b - y)$.*

- *Suppose we follow the policy that sets $\lambda_i = \lambda_m^{\text{Kelly}}$ for all $i \in \{t+1, \ldots, N\}$. Then, we have*

$$\Delta \geq B\sqrt{8T \log T} \implies \mathbb{P}(\tau \leq N) \geq 1 - \frac{1}{T}.$$

- *Now, let us consider any other policy that plays $\{\lambda_{t+1}, \ldots, \lambda_N\}$ taking values in $\Lambda$ such that for some $\rho > 0$, the condition $\big| \{i \in \{t+1, \ldots, t+k\} : |\lambda_i - \lambda_m^{\text{Kelly}}| \geq \delta\} \big| \geq \rho k$ holds for all $k \in \{1, 2, \ldots, T\}$. Then, we have*

$$\Delta \leq \rho\epsilon T - B\sqrt{8T \log 2} \implies \mathbb{P}(\tau \leq N) \leq \frac{1}{2}.$$

We prove this result in Appendix B.1.

*Remark* 3.2. Theorem 3.1 formalizes the phenomena that in the interior of the $(t, \log W_t)$ plane, policies that deviate significantly from Kelly betting cannot be optimal. In particular, with $T = N - t$ denoting the time-to-go, and $\Delta = TL_{\max} - (b - y)$, the first part of the result says that if $\Delta \geq B\sqrt{8T \log T}$, then the Kelly betting policy rejects the null with probability at least $1 - 1/T$. In other words, if the bettor is almost "on-schedule", with the required per-step drift $(b - y)/T$ approximately $L_{\max} - \sqrt{\log T/T}$, then the Kelly policy will reliably hit the required threshold with high probability especially when $T$ is large. On the other hand, any policy that spends a nontrivial fraction $\rho$ of the remaining time playing bets that are $\delta$-away from Kelly suffers a linear drift deficit of the order $\rho\epsilon T$. To hit the threshold with such a policy, the wealth process needs an upward martingale fluctuation of at least the size $\rho\epsilon T - \Delta$. The second part of the result says that if $\Delta \leq \rho\epsilon T - B\sqrt{8T \log 2}$, then this event happens with probability at most $1/2$. To summarize, in this "tubular" interior region of the state space with $\sqrt{T \log T} \lesssim \Delta \lesssim \rho\epsilon T - \sqrt{T}$, any near-optimal strategy must stay close to Kelly for large portions of the remaining time.

*Example* 3.3. We now discuss an example in which the conditions of Theorem 3.1 are satisfied. Fix $m = 1/2$, and $X \sim \text{Bernoulli}(p)$ with $p = 0.7$. The Kelly bet in this case is $\lambda_m^{\text{Kelly}} = 4p - 2 = 0.8$ with $L_{\max} \approx 0.082$. If we fix $\Lambda = [0, 1]$, then we get $B = \log 2 \approx 0.69$. Moreover, if we select $\delta = 0.6$, then we get $\epsilon \equiv \epsilon(\delta) = L(0.8) - L(0.2) \approx 0.047$. Finally, consider a state $(t, y)$ with $T = N - t = 50000$, and $b - y = 1475$. Then, observe that $B\sqrt{8T \log T} \approx 1442 < \Delta$, which means that the Kelly strategy hits the threshold with probability at least $1 - 1/T = 1 - 2 \times 10^{-t}$. Moreover, if $\rho = 0.8$, then we have $\rho\epsilon T - B\sqrt{8T \log 2} \approx 1532 > \Delta$, which means that any policy that deviates a fraction $\rho$ of time from Kelly must have $\mathbb{P}(\tau \leq N) \leq 1/2$.

The previous result suggests that good policies cannot de-

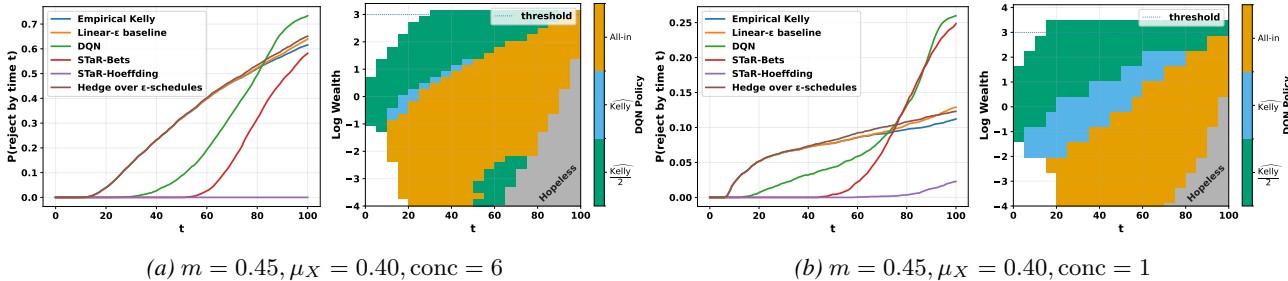

*(a)* $m = 0.45, \mu_X = 0.40, \text{conc} = 6$       *(b)* $m = 0.45, \mu_X = 0.40, \text{conc} = 1$

*Figure 3.* We compare the probability of rejection by time $t$ with $N = 100$, $\alpha = 0.05$, and null mean $m = 0.45$. We simulate 5,000 length-$N$ sequences $\{X_i\}$ from a two-component Beta mixture. We define the *hopeless* region as the set of states where the optimal policy attains $\mathbb{P}(\text{reject by } N) < 10^{-4}$. In each of **(a)** and **(b)**, the right panel shows the DQN's most frequently selected action, illustrating the learned phase diagram adapts across data-generating distributions. Linear-$\epsilon$ baseline is implemented with $\eta = 1/2$ and the Hedge over $\epsilon$ is implemented with varying $\eta$. All-in refers to $\lambda_{\max}$. The details are provided in Section 4.2. **(a):** $X_i \sim \text{Beta}(2.4, 3.6)$ with probability $1/2$ and $X_i \sim \text{Beta}(4.8, 7.2)$ with probability $1/2$. **(b):** $X_i \sim \text{Beta}(0.4, 0.6)$ w.p. $1/2$ and $X_i \sim \text{Beta}(0.8, 1.2)$ w.p. $1/2$.

viate too much too often from Kelly betting certain "interior" regions of the $(t, \log W_t)$ plane. We now present two results that give us sufficient conditions under which it is advantageous to deviate from Kelly betting. To simplify the arguments, we will focus on the case of distributions with finite support $\mathcal{X} \subset [0,1]$ with $|\mathcal{X}| = k$. First we consider an example, in which the bettor has "fallen behind", which formally means the following, with $B_K := \max_{x \in \mathcal{X}} h_m(\lambda_m^{\text{Kelly}}, x)$:

$$r \equiv r(t, y, b) := \frac{b - y}{N - t} > \max\left\{L_{\max}, \frac{1}{2}B_K\right\}. \quad (3)$$

In words, this condition says that the average drift $(b - y)/(N - t)$ needed to hit the threshold in $T = N - t$ steps is larger than the one-step drift in log-wealth induced by the Kelly bet, $L_{\max} = L(\lambda_m^{\text{Kelly}})$. Additionally, we also impose the condition that it is impossible for the bettor to hit the threshold in fewer than $T/2$ steps using Kelly bets. Now, for each $\lambda$, define the following KL-rate function, with $D$ denoting the KL divergence (or relative entropy):

$$I^+(\lambda, r) = \inf_{Q \in \mathcal{Q}^+(r, m, \lambda)} D(Q \| P_X), \quad \text{where}$$
$$\mathcal{Q}^+(r, m, \lambda) = \left\{Q \in \mathcal{P}(\mathcal{X}) : \mathbb{E}_Q[h_m(\lambda, X)] \geq r\right\}. \quad (4)$$

Thus, $I^+(\lambda, r)$ denotes the minimum KL divergence between a sufficiently tilted distribution $Q$ that allows the drift to exceed $r$, and the true distribution $P_X$ under consideration. We now state a precise condition under which a more aggressive (than Kelly) betting policy has a higher probability of hitting the threshold.

**Proposition 3.4.** *Suppose $(t, y)$ satisfy the condition in (3), and let $\tau(\lambda)$ denote the stopping time associated with the policy that plays a constant bet $\lambda$ for all $T = N - t$ steps starting at $(t, y)$. Then, for any $\lambda^{\text{agg}} > \lambda_m^{\text{Kelly}}$, we have*

$$I^+(\lambda^{\text{agg}}, r) < \frac{I^+(\lambda_m^{\text{Kelly}}, r)}{2} - c_T^+$$
$$\implies \mathbb{P}(\tau(\lambda_m^{\text{Kelly}}) \leq N) \leq \mathbb{P}(\tau(\lambda^{\text{agg}}) \leq N).$$

*where the term $c_T^+ = \mathcal{O}\left(\frac{\log T}{T}\right)$ is defined explicitly in* (11) *in Appendix B.2.*

The proof of this result is in Appendix B.2. The proposition above provides a rigorous though conservative finite-$T$ sufficient condition, since it relies on combinatorial type-based large-deviation arguments. To build intuition, we now present an example, that suppresses the finite-$T$ correction term $(c_T^+)$ and compares the two rate function terms $I^+(\lambda^{\text{agg}}, r)$ and $(1/2)I^+(\lambda^{\text{Kelly}}, r)$ directly. We then verify by direct calculation that the corresponding hitting probabilities are also ordered in the same way.

*Example* 3.5. Fix $m = 1/2$ and $X \sim \text{Bernoulli}(p)$ with $p = 0.6$. Take $\alpha = 0.05$, so $b = \log(1/\alpha) = \log 20$. Suppose $T = N - t = 20$ steps remain and $y = \log W_t = 0$, so $r = (b - y)/T = \log 20/20 \approx 0.150$. The Kelly bet is $\lambda_m^{\text{Kelly}} = \frac{p - m}{m(1 - m)} = 0.4$ with $L_{\max} = L(\lambda_m^{\text{Kelly}}) \approx 0.020$ and $B_K = \max_{x \in \{0,1\}} h_m(\lambda_m^{\text{Kelly}}, x) = \log(1.2) \approx 0.182$, hence $r > \max\{L_{\max}, \frac{1}{2}B_K\}$ and we are in (3). Moreover $1.2^{10} < 20$, so Kelly cannot reach the threshold within $T/2$ steps. Taking instead an aggressive bet $\lambda^{\text{agg}} = 1.5$ gives multipliers $1.75$ on $X = 1$ and $0.25$ on $X = 0$, so $1.75^6 > 20$, suggesting that $\lambda^{\text{agg}}$ might be a better option. Indeed, direct calculation reveals that $I^+(\lambda^{\text{agg}}, r) \approx 0.081 < (1/2)I^+(\lambda^{\text{Kelly}}, r) \approx 0.132$, and $\mathbb{P}(\tau(\lambda^{\text{agg}}) \leq N) \approx 0.13$ beats $\mathbb{P}(\tau(\lambda_m^{\text{Kelly}}) \leq N) \approx 8.6 \times 10^{-4}$.

We now consider the complementary situation in which the bettor is ahead of schedule and thus it may be advantageous to bet less aggressively than even Kelly betting. Formally, we consider the state $(t, y)$ such that with $T = N - t$,

$$\frac{1}{2}B_K < r \equiv r(t, y, b) = \frac{b - y}{T} < L_{\max}, \quad (5)$$

with $B_K = \max_{x \in \mathcal{X}} h_m(\lambda_m^{\text{Kelly}}, x)$ as before. The left inequality ensures that even Kelly betting needs more than $T/2$ steps to hit the threshold, while the right inequality formalizes the meaning of "ahead of schedule" in the sense

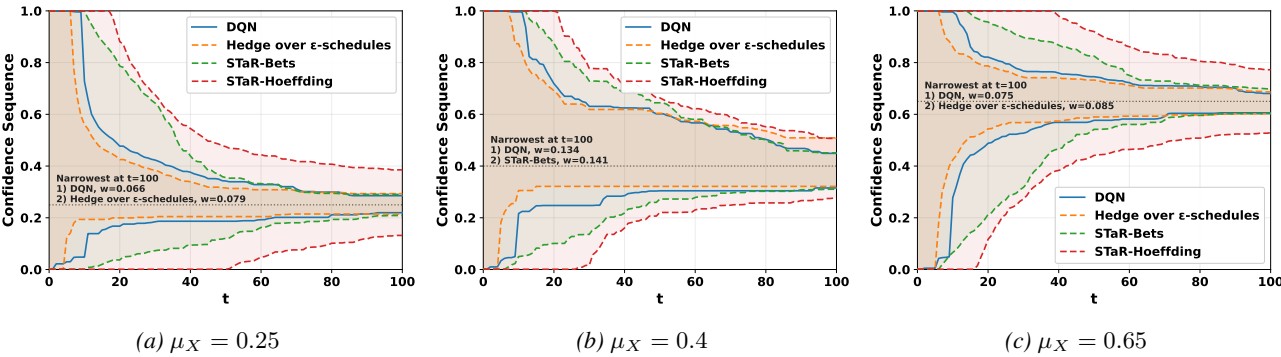

*Figure 4.* We compare confidence sequences under different data-generating distributions with $\alpha = 0.05$ following Section 3.1. DQN produces the narrowest confidence intervals at deadline $t = N = 100$ and also yields tight intervals in the early steps. **(a)** $X_i \sim \text{Beta}(2, 6)$ w.p. 1/2 and $X_i \sim \text{Beta}(4, 12)$ w.p. 1/2. **(b)** $X_i \sim \text{Beta}(0.4, 0.6)$ w.p. 1/2 and $X_i \sim \text{Beta}(0.8, 1.2)$ w.p. 1/2. **(c)** $X_i \sim \text{Beta}(6.5, 3.5)$ w.p. 1/2 and $X_i \sim \text{Beta}(13, 7)$ w.p. 1/2.

that the average drift needed, $r$, is strictly less than the Kelly drift $L_{\max}$. Next, introduce the rate function

$$I^-(\lambda, r) := \inf_{Q \in \mathcal{Q}^-(r, m, \lambda)} D(Q \| P_X), \quad \text{where}$$
$$\mathcal{Q}^-(r, m, \lambda) = \{Q \in \mathcal{P}(\mathcal{X}) : \mathbb{E}_Q[h_m(\lambda, X)] \le r\}. \tag{6}$$

We can now state a result formalizing the sufficient conditions for a defensive betting strategy to be better than Kelly.

**Proposition 3.6.** *Suppose $(t, y)$ satisfy (5), and let $\tau(\lambda)$ denote the stopping time associated with the policy that plays a constant bet $\lambda$ for all $T = N - t$ steps starting at $(t, y)$. Then, with $r_- := 2\left(r - \frac{1}{2}\max_{x \in \mathcal{X}} h_m(\lambda_m^{\text{Kelly}}, x)\right)$, for any feasible $\lambda^{\text{def}} < \lambda_m^{\text{Kelly}}$, we have*

$$I^-(\lambda^{\text{def}}, r) > \frac{1}{2}I^-(\lambda_m^{\text{Kelly}}, r_-) + c_T^-(\lambda^{\text{def}}, r)$$
$$\implies \mathbb{P}(\tau(\lambda^{\text{def}}) \le N) \ge \mathbb{P}(\tau(\lambda_m^{\text{Kelly}}) \le N),$$

*where $c_T^-(\lambda^{\text{def}}, r) = \mathcal{O}\left(\frac{\log T}{T}\right)$ is defined explicitly in (13) in Appendix B.3.*

The proof of this result follows the same template as that of Proposition 3.4, and we present the details in Appendix B.3.

*Example* 3.7. To illustrate Proposition 3.6, consider $X \sim \text{Bernoulli}(p)$ with $p = 0.99$, $m = 0.2$, and let $\alpha = 0.05$, so that $b = \log(20)$ as before. Consider a state with $T = N - t = 5$, and $y = b - rT$ with $r = 0.8$; so $y \approx -1.0$. The Kelly bet in this case is 4.94, with one-step maximum increment of $B_K \approx 1.60$ and drift $L_{\max} \approx 1.54$. Hence, in this case we have $\frac{1}{2}B_K < r < L_{\max}$, and the condition (5) holds. Let us define $r_- = 2(r - \frac{1}{2}B_K) \approx 6 \times 10^{-4}$, and select a defensive bet $\lambda^{\text{def}} = 0.75\lambda_m^{\text{Kelly}} \approx 3.70$. A direct computation shows that $I^-(\lambda^{\text{def}}, r) \approx 0.466 > (1/2)I^-(\lambda_m^{\text{Kelly}}, r_-) \approx 0.329$,

ignoring the finite $T$ correction term. So the condition required by Proposition 3.6 holds (ignoring the finite-$T$ correction term $c_T^-$), and a direct calculation shows that we have $\mathbb{P}(\tau(\lambda^{\text{def}}) \le N) \approx 0.999 \ge \mathbb{P}(\tau(\lambda_m^{\text{Kelly}}) \le N) \approx 0.970$.

### 3.1. Horizon-Aware Confidence Sequences

Once we know how to construct power-one tests, we can use them to define horizon-aware time-uniform confidence sequences $\{C_n : 0 \le n \le N\}$ with $C_0 = [0, 1]$, and

$$C_n = \{m \in [0, 1] : \tau_m > n\}$$
$$= \{m \in [0, 1] : W_n(m) < 1/\alpha\}, \forall n \in \{1, \dots, N\}.$$

Since each $\tau_m$ is level-$\alpha$ by construction, it follows that

$$\mathbb{P}\left(\exists n \ge 1 : \mu_X \notin C_n\right) \le \mathbb{P}\left(\tau_{\mu_X} < \infty\right) \le \alpha,$$

where $\mu_X$ denotes the true and unknown mean. Thus, constructing a confidence sequence can be considered as running in a continuum of hypothesis tests in parallel, and retaining those values of $m$ for which the tests have not rejected the null $H_{0,m}$ yet.

## 4. From Phase Diagram to Policies: Heuristics and Deep Q-Learning

The theory above provides a partial characterization of the optimal betting policy in the horizon-aware setting, which we summarize in Figure 1. In short, for $(t, y)$ in a "tubular" region in the interior of the state space, where hitting the threshold does not require atypical fluctuations, any near-optimal policy cannot deviate too much from Kelly betting (Theorem 3.1). In contrast, Proposition 3.4 considers a "behind schedule" scenario, where threshold crossing itself is a large deviation event, and in this case a more aggressive bet can increase the probability of rejection. Finally, Proposition 3.6 identifies an "ahead of schedule" regime

in which a more defensive betting strategy can reduce the chances of large downward jumps in the wealth that cannot be recovered from before the deadline. These three regimes align well with the representative phase diagram of Figure 1.

Since the true Kelly bet depends on the unknown distribution $P_X$, we instead use a predictable empirical Kelly estimate $\widehat{\lambda}_t(m)$ in our experiments. The goal is then to bet less aggressively, approximately Kelly, or more aggressively depending on where we are relative to schedule. The central challenge is determining this schedule position in practice, since we do not know the true distribution. Moreover, the phase diagram itself changes with problem difficulty. For example, how far the null $m$ is from the true mean $\mu_X$, and with distributional properties of $\{X_i\}$ such as variance and tail behavior. To build intuition about how optimal betting policies vary across the $(t, \log W_t)$ plane as difficulty changes, we compute the exact optimal action as a function of the current state $(t, \log W_t)$.

### 4.1. Phase Diagram of Optimal Actions

Given a finite action set at each state $(t, \log W_t)$, we use the Bellman recursion in Equation 2 to compute (i) the value at each $(t, \log W_t)$ point and (ii) the optimal action that attains this value. We initialize the dynamic program at $t = N - 1$, where $N$ is the deadline, and then compute the value function for all earlier times $t < N - 1$ via backward induction. We consider three actions: (All-in, $\lambda_m^{\text{Kelly}}$, $\lambda_m^{\text{Kelly}}/2$). All-in refers to $\lambda_{\max}$. We discretize the $(t, \log W_t)$ plane into an evenly spaced grid and, at each grid point, compute the optimal action using the Bellman recursion. Note that evaluating the Bellman recursion requires knowledge of $P_X$. Therefore, we use the population Kelly bet.

Figure 2 shows results for $X_i$ drawn from a two-component Beta mixture: $X_i \sim \text{Beta}(2.4, 3.6)$ with probability $1/2$ and $X_i \sim \text{Beta}(4.8, 7.2)$ with probability $1/2$, which has mean $\mu_X = 0.40$ and time-varying variance across realizations. We set the deadline to $N = 100$ and the significance level to $\alpha = 0.05$. We compute the phase diagram for the null means $m = 0.45$, $m = 0.43$, and $m = 0.41$.

In Figure 2a, the optimal action follows Kelly in a central band (i.e., when the process is "on schedule"). Once it moves ahead of schedule, the optimal action becomes fractional Kelly, whereas when it falls behind schedule, the optimal action switches to All-in. As the problem becomes more difficult (e.g., smaller $m = 0.43$ in Figure 2b), the All-in and Kelly regions shift upward, and the optimal action space favors more aggressive bets. When $m = 0.41$ in Figure 2c, the conservative fractional-Kelly region disappears, the Kelly region narrows, and the All-in region widens. Overall, as difficulty increases, the optimal policy favors increasingly aggressive actions.

Figure 2 highlights a key practical challenge in identifying the optimal action regions. The phase diagram is dynamic: as problem difficulty changes, the Kelly, fractional-Kelly, and aggressive-betting bands shift. Moreover, the phase diagram depends on the data-generating distribution $P_X$, which is unknown in practice. Finally, in applications we rely on a predictable empirical Kelly estimate $\widehat{\lambda}_t(m)$, which introduces estimation uncertainty. In particular, aggressive (or even approximately Kelly) actions early on, when $t$ is small and $\widehat{\lambda}_t(m)$ is most uncertain, may be undesirable.

We now discuss how to design schedule-aware betting policies. We first propose a simple schedule-aware heuristic, which serves as a strong baseline. We then describe how to learn schedule-aware and distributionally sensitive betting policies using Deep Q-learning.

### 4.2. $\epsilon$-Greedy Schedules

A simple schedule-aware heuristic is an $\epsilon$-greedy *mixed-strike* rule: at time $t$, play the empirical Kelly bet $\widehat{\lambda}_t(m)$ with probability $1 - \epsilon_t$, and an aggressive bet (i.e., $\lambda_{\max}$) with probability $\epsilon_t$. Here $\epsilon_t \in [0, 1]$ is a monotone nondecreasing function of $t$ with $\epsilon_t \uparrow 1$ as $t \uparrow N$, so the policy becomes increasingly aggressive as the deadline approaches.

We consider two schedules for $\epsilon_t$: linear and quadratic growth. To delay aggressiveness until late in the horizon, we introduce an *aggressiveness onset fraction* $\eta \in \{0.25, 0.50, 0.75\}$. We set $\epsilon_t = 0$ for all $t < \eta N$, and for $t \in [\eta N, N]$ we increase $\epsilon_t$ according to the chosen trend, normalized so that $\epsilon_{\eta N} = 0$ and $\epsilon_N = 1$.

**Hedging over $\epsilon$-schedules.** The mixed-strike heuristic depends on the choice of the exploration schedule $\{\epsilon_t\}_{t=1}^N$, and different schedules can perform best under different distributions $P_X$ and null values $m$. We therefore hedge uniformly over a set of $K = 6$ schedules described above, following the e-process hedging principle of Waudby-Smith & Ramdas (2024). We provide the details in Appendix C.

### 4.3. Deep-Q-Learning

Realize that horizon-aware betting is a finite-horizon sequential decision problem. At each time $t \leq N$, we choose a predictable bet $\lambda_t(m) \in \Lambda_m$, update wealth as $W_t = W_{t-1}\big(1 + \lambda_t(m)(X_t - m)\big)$ and aim to maximize the probability of crossing the threshold $\log W_t \geq \log(1/\alpha)$ by time $N$. The Bellman recursion in (2) casts this as a dynamic program over $(t, \log W_t)$, with nonstationarity driven by the remaining time $N - t$ and the gap $\log(1/\alpha) - \log W_t$. In practice, the optimal policy depends on the unknown $P_X$, so the phase diagram shifts with $m$ and distributional properties that must be learned online.

Reinforcement learning is natural here as it optimizes finite-

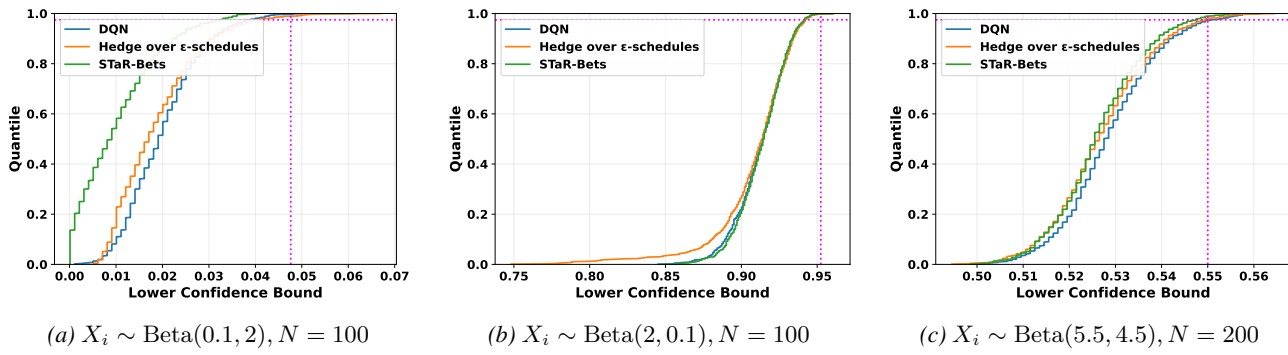

*(a)* $X_i \sim \mathrm{Beta}(0.1, 2), N = 100$    *(b)* $X_i \sim \mathrm{Beta}(2, 0.1), N = 100$    *(c)* $X_i \sim \mathrm{Beta}(5.5, 4.5), N = 200$

*Figure 5.* We plot the lower confidence bounds across different Beta distributions and deadlines $N$. When a method's curve passes through a point $(x, y)$, it means that over 5000 repetitions, the lower confidence bound was below $x$ in a fraction $y$ of the runs. The vertical line indicates the mean of the distribution, and the horizontal line indicates the $(1 - \alpha/2)$ quantile. Lower curves indicate better performance across methods. Note that in all cases, at $x = \mu_X$ we observe $y > 1 - \alpha/2$, consistent with the validity of the methods.

horizon control under uncertainty without an explicit model. We treat each test run as an episode of length at most $N$, terminating upon threshold crossing, and use the sparse terminal reward

$$R = \mathbf{1}\left\{ \max_{1 \le t \le N} \log W_t \ge \log(1/\alpha) \right\} \qquad (7)$$

Then $\mathbb{E}[R] = \mathbb{P}(\tau_m \le N)$, so maximizing return maximizes deadline-limited power. Anytime-validity is unchanged, since it depends only on predictability and the constraint $\lambda_t(m) \in \Lambda_m$, not on how the policy is learned.

We use Deep Q-Learning with a small, interpretable discrete action set that matches the phase-diagram intuition: $\widehat{\lambda}_t(m)/2$, $\widehat{\lambda}_t(m)$, and $\lambda_{\max}$. Q-learning (Watkins & Dayan, 1992) learns an action-value function $Q(s, a)$, and the Deep Q-Network (DQN) generalizes this across the continuous state space (Mnih et al., 2015). This is well-suited to our sparse-reward, finite-horizon and continuous state setting as the learned $Q$-function aims to yield a greedy policy that selects the action with highest estimated success probability.

Although the DP state is $(t, \log W_t)$ for fixed $m$ and $P_X$, the data-generating distribution is unknown in practice. We therefore provide the DQN with a predictable feature vector computed only from $\mathcal{F}_{t-1}$ (e.g., empirical moments, time remaining, distance-to-threshold, $m$, and empirical Kelly/endpoint proxies), and map actions to bets clipped into $\Lambda_m$. The resulting policy is *schedule-aware* (adapting as $t \uparrow N$ and the gap changes) and *distribution-sensitive* (reacting differently across distributional regimes), while remaining compatible with anytime-valid inference.

Under this setting, we train a *universal* DQN policy using synthetic episodes over randomized $N, m$ and a family of worlds (i.e., Beta and Beta-mixtures with randomized parameters), so a single policy generalizes across horizons, distributions, and null values. This addresses the main issue suggested by the phase diagram: the conservative, Kelly,

and aggressive regions are not fixed, but shift with problem difficulty and unknown distributional properties. Note that DQN is trained only once over 500,000 of episodes of random distributions, null values and deadlines. We expand on the implementation details in Appendix E.

### 4.4. Numerical Results

To the best of our knowledge, Voráček & Orabona (2025) is the only prior work to construct horizon-aware, betting-based confidence intervals and tests, proposing the STaR-Bets and STaR-Hoeffding methods. We therefore compare our horizon-aware strategies with these methods, as well as with horizon-agnostic baselines such as predictable empirical Kelly betting. Voráček & Orabona (2025) focuses on one-sided tests. We adapt them to the two-sided case while preserving the core principles of STaR-Bets and STaR-Hoeffding. Implementation details are in Appendix D.

**Deadline-limited power and learned phase diagrams.** Figure 3 considers $N = 100$ testing with $m = 0.45$ under two Beta-mixture worlds that share mean $\mu_X = 0.40$ but differ in concentration. In both regimes, the DQN increases the probability of rejecting by the deadline. The modal-action maps (right panels) recover the qualitative structure suggested by our theory and the oracle phase diagrams in Figure 2: a central Kelly-preferring band when on schedule, a conservative region when ahead, and a switch to All-in when behind and/or when $N - t$ is small. Early in the horizon, the policy is more conservative, consistent with higher estimation uncertainty in $\widehat{\lambda}_t(m)$. Importantly, these action regions shift across the two worlds, indicating distribution-sensitive adaptation rather than a fixed schedule.

This picture is supported by the trajectory-level visualizations in Appendix F.9. Across choices of $N$, $m$, and distributional regime, the same three-region structure persists, and the regime boundaries and phase-shift times vary with problem difficulty. Along individual sample paths, the learned

policy switches between $\widehat{\lambda}_t(m)/2$, $\widehat{\lambda}_t(m)$, and $\lambda_{\max}$ in response to realized wealth and time remaining: when a path drifts sufficiently below the on-schedule region or the deadline becomes tight, the policy often enters sustained aggressive phases, while after recovery it typically returns to Kelly or half-Kelly. In easier worlds, these aggressive episodes are shorter and less frequent, whereas in harder or more diffuse worlds, they last longer. Together, the modal-action maps and pathwise plots indicate that the DQN is implementimng a state-dependent and distribution-sensitive betting policy.

**Horizon-aware confidence sequences.** Figure 4 compares confidence sequences for $N = 100$ across three mixture settings. DQN yields the narrowest confidence sets at the deadline $t = N$ and also remains tight in early steps, indicating that optimizing finite-horizon rejection probability also sharpens earlier estimation. Figure 5 provides a complementary analysis via the empirical CDF of lower confidence bounds under several Beta distributions and deadlines ($N \in \{100, 200\}$): DQN produces lower curves (tighter lower bounds) while remaining valid at $x = \mu_X$.

**Alternative reward functions.** Our control framework can steer the learned betting policy by changing only the reward function. Figure 6 compares the original terminal-power objective with two early-rejection variants. We train two reward-modified DQN variants: DQN-EB with early-bonus reward $r = 1 + (1 - t/N)$, and DQN-U with urgency reward $r = 1 - t/N$, and compare them to the original DQN reward given by Equation 7) and existing baselines. Figure 6 illustrates the expected tradeoff: the original DQN achieves the strongest terminal power, DQN-EB gives a middle ground, and DQN-U shifts rejection earlier at some cost in power at $t = N$. These variants may be preferable when earlier rejection is more important, for example to obtain narrower confidence sequences well before the deadline.

**Generalization to other distributions.** Figure 7 shows that a policy trained on the Beta family in this work also transfers to out-of-distribution logit-normal data, where it continues to improve power over existing baselines. This suggests that Beta-trained policies can generalize beyond the training distribution family. In Figure 8, we further evaluate the same control/RL framework on Bernoulli families, training across $\mu_X, m \in [0, 1]$ and testing at $\mu_X \in \{0.10, 0.30, 0.50\}$ with $\alpha = 0.05$. It again improves power over baselines, suggesting that the approach is not tied to continuous distributions.

**Real data experiments.** We evaluate whether a DQN policy trained only on synthetic Beta and Beta-mixture distributions transfers to real data. We consider two real-data domains comprising six unseen data sources: genome-wide DNA methylation beta values and daily relative humidity measurements. In each case, we draw $N = 100$ observations with replacement from the empirical distribution, run the sequential test $H_0 : \mu = m$ at level $\alpha = 0.05$, and

repeat this for 5,000 trials to estimate $\mathbb{P}(\text{reject by time } t)$. The DQN policy is not fine-tuned on real data. Full dataset descriptions, histograms, rejection curves, and Type-I error checks are provided in Appendix F.3; see Figures 9–11 for DNA methylation and Figures 12–14 for relative humidity.

Across these six real-data settings, DQN achieves the highest rejection probability by the deadline in 5/6 cases and is very close to the best method in the remaining case, while maintaining valid null calibration throughout. These results provide evidence that the gains are not limited to synthetic settings: the learned policy transfers without retraining to real distributions from two distinct application domains, reflecting a more general horizon-aware betting strategy.

**Further experimental results.** We first ablate whether a larger action space improves performance. In Figure 15, we train an additional DQN policy with an expanded 9-action space. We observe that the larger action space yields similar performance but does not lead to an overall improvement. Therefore, for simplicity, we adopt the 3-action policy in the main paper. Second, we evaluate the same DQN policy used in the paper at longer deadlines, namely $N \in \{250, 300, 350\}$. Figure 16 shows that DQN maintains improved power under these extended horizons. Third, we show that DQN can accommodate different $\alpha$ values. We set $\alpha = 0.01$ and retrain a DQN policy for this level. Figure 17 shows that DQN also improves power at $\alpha = 0.01$.

## 5. Conclusion

We study *horizon-aware, anytime-valid* tests and confidence sequences under a hard deadline $N$. Viewing horizon-aware betting as a finite-horizon control problem over $(t, \log W_t)$ makes explicit how optimal actions depend on time remaining and distance to the threshold. This leads to a three-regime phase diagram: when on schedule, substantial deviations from Kelly are provably suboptimal (Theorem 3.1). When behind schedule, aggressive bets can increase the probability of crossing by $N$ (Proposition 3.4). When ahead of schedule, defensive bets mitigate unrecoverable drawdowns (Proposition 3.6). Oracle DP phase diagrams validate this structure and show that regime boundaries shift with difficulty and distributional properties. A universally trained DQN recovers a similar partition in modal-action maps across distributions, and improves deadline-limited power and tightens confidence sequences while preserving validity by construction. Hedging over $\epsilon$-greedy schedules is strong, but the DQN typically performs better by leveraging richer predictable features to switch state-dependently among fractional-Kelly, Kelly, and all-in actions.

Our experiments focus mainly on continuous distributions. Extending DQN to more general settings, such as mixed distributions, is an intriguing direction for future work.

## Acknowledgments

This work is supported by the National Science Foundation grants CCF-2046816, CCF-2403075, CCF-2212426, the Office of Naval Research grant N000142412289, a gift from Open Philanthropy, and an Adobe Data Science Research Award. The computational aspects of the research are generously supported by computational resources provided by the Amazon Research Award on Foundation Model Development.

## Impact Statement

This paper formalizes horizon-aware anytime-valid inference from a finite-horizon control perspective and develops learning-based betting policies for this setting. Hypothesis testing is a backbone of scientific inference, and anytime-valid methods are particularly useful in applications such as online A/B testing, adaptive experimentation, and resource-constrained scientific studies, where analysts may monitor data continuously while facing a fixed deadline. By improving deadline-limited power and tightening confidence sequences while preserving validity, our approach can make sequential inference more effective in practical deployments. A potential limitation is that learning-based policies are generally less interpretable than hand-crafted testing rules. In settings where transparency, auditability, or simple human-understandable decision rules are essential, this reduced interpretability may present a practical concern.

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

# A. Additional Background

In this section, we recall some technical results that we use throughout the paper. The first result is a time-uniform generalization of Markov's inequality, due to Ville (1939), that is applicable to nonnegative supermartingales.

**Fact A.1.** Ville's Inequality Suppose $\{W_n : n \geq 0\}$ denotes a nonnegative martingale adapted to a filtration $\{\mathcal{F}_n : n \geq 0\}$ with $W_0 = 1$ (a.s.). Then, for any $\alpha \in (0, 1]$, we have

$$\mathbb{P}\left(\exists n \geq 1 : W_n \geq \frac{1}{\alpha}\right) \leq \alpha \quad \Longleftrightarrow \quad \mathbb{P}\left(\forall n \geq 1 : W_n < \frac{1}{\alpha}\right) \geq 1 - \alpha.$$

Next, we recall a large deviation inequality for the case of random variables taking values in a finite alphabet $\mathcal{X}$. The specific form we present here is from Cover & Thomas (2006, Theorem 11.4.1).

**Fact A.2** (Sanov's Theorem for finite alphabets). Let $X_1, \ldots, X_n \overset{\text{i.i.d.}}{\sim} P_X$ denote $n$ $\mathcal{X}$-valued random variables with empirical distribution $\widehat{P}_n$, and $\mathcal{P}_n(\mathcal{X})$ denote the set of all types or empirical distributions with $n$ observations on $\mathcal{X}$. Then, for any set $E \subset \mathcal{P}(\mathcal{X})$ that is the closure of its interior and with $E_n = E \cap \mathcal{P}_n(\mathcal{X})$, we have

$$\frac{1}{(n+1)^{|\mathcal{X}|}} e^{-nD(Q^*(E_n)\|P_X)} \leq \mathbb{P}_{X^n}\left(\widehat{P}_n \in E\right) \leq (n+1)^{|\mathcal{X}|} e^{-nD(Q^*(E)\|P_X)},$$

where $Q^*(E) = \operatorname{argmin}_{Q \in E} D(Q \parallel P_X)$, and $Q^*(E_n) = \operatorname{argmin}_{Q \in E_n} D(Q \parallel P_X)$.

We now state and prove a simple technical result that will be used in the proof of Proposition 3.4 and Proposition 3.6.

**Lemma A.3.** *Let $\mathcal{X} \subset [0, 1]$ denote a finite alphabet of size $k$, and let $\mathcal{P}_n(\mathcal{X})$ denote the set of all "types" on $\mathcal{X}$ with $n$ observations; that is,*

$$\mathcal{P}_n(\mathcal{X}) = \left\{\frac{1}{n}\sum_{i=1}^{n} \delta_{x_i} : (x_1, \ldots, x_n) \in \mathcal{X}^n, \ n \geq 1\right\},$$

*where $\delta_{x_i}$ denotes the point mass at $x_i$. Fix a distribution $P_X \in \mathcal{P}(\mathcal{X})$ with $p_{\min} = \min_{x \in \mathcal{X}} P_X(x) > 0$. For an $m \in [0, 1]$ and $\lambda \in \Lambda_m$, let $h_m(\lambda, x) = \log(1 + \lambda(x - m))$ and define*

$$I_n^+(\lambda, r) = \inf\left\{D(Q \parallel P_X) : Q \in \mathcal{P}_n(\mathcal{X}), \ \mathbb{E}_Q[h_m(\lambda, X)] \geq r\right\},$$

$$\text{and} \quad I_n^-(\lambda, r) = \inf\left\{D(Q \parallel P_X) : Q \in \mathcal{P}_n(\mathcal{X}), \ \mathbb{E}_Q[h_m(\lambda, X)] \leq r\right\}.$$

*With $I^+$ and $I^-$ as defined in (4) and (6) and $C \equiv C(n, k, p_{\min}) = \left(2 + \log\left(\frac{n}{p_{\min}}\right)\right)\frac{k}{n}$, we have*

$$I_n^+(\lambda, r) \leq I^+(\lambda, r) + C(n, k, p_{\min}), \quad \text{and} \quad I_n^-(\lambda, r) \leq I^-(\lambda, r) + C(n, k, p_{\min}).$$

*Proof.* We will prove the statement for $I^+$ since an exactly analogous argument also implies the result for $I^-$. Let $Q^* \equiv (q_1^*, \ldots, q_k^*)$ denote the distribution achieving the infimum in $I^+(\lambda, r)$. The existence of such a $Q^*$ is guaranteed since the domain of optimization is a closed subset of the simplex. For each $i \in [k]$, introduce

$$f_i := nq_i^* - \lfloor nq_i^* \rfloor, \quad \text{and} \quad R = \sum_{i=1}^{k} f_i \in \{0, \ldots, k-1\}.$$

Now, let $i_1, i_2, \ldots, i_k$ denote a permutation of the elements of $[k]$, such that $h_m(\lambda, x_{i_1}) \geq h_m(\lambda, x_{i_2}) \geq \ldots \geq h_m(\lambda, x_{i_k})$. Then, we can define a new distribution in $\mathcal{P}_n(\mathcal{X})$ by perturbing $Q^*$ as

$$Q_n \equiv (q_1, \ldots, q_k), \quad \text{with} \quad q_i = \begin{cases} \frac{\lfloor nq_i^* \rfloor + 1}{n}, & \text{if } i \in \{i_1, \ldots, i_R\}, \\ \frac{\lfloor nq_i^* \rfloor}{n}, & \text{otherwise.} \end{cases}$$

In other words, $Q_n$ is a quantized version of $Q^*$ lying in $\mathcal{P}_n(\mathcal{X})$ which is biased to have a larger value of $E_{Q_n}[h_m(\lambda, X)]$, and thus remains feasible for $I^+(\lambda, r)$. In particular, we have

$$|q_i - q_i^*| \leq \frac{1}{n} \text{ for all } i \in [k], \quad \text{and} \quad \sum_{i=1}^{k}(q_i - q_i^*)h_m(\lambda, x_i) \geq 0.$$

Then, with $\phi(u) = u \log u$ for $u > 0$ and $\phi(0) = 0$, we have

$$
\begin{aligned}
D(Q_n \parallel P_X) - D(Q^* \parallel P_X) &= \sum_{i=1}^{k} \phi(q_i) - \phi(q_i^*) + \sum_{i=1}^{k} (q_i - q_i^*) \log \left( \frac{1}{P_X(x_i)} \right) \\
&\leq \sum_{i=1}^{k} \phi(q_i) - \phi(q_i^*) + \frac{k}{n} \log \left( \frac{1}{p_{\min}} \right) \\
&\leq \frac{k(2 + \log n)}{n} + \frac{k}{n} \log \left( \frac{1}{p_{\min}} \right).
\end{aligned}
$$

On taking infimum over $Q_n$ this implies the required $I_n^+(\lambda, r) \leq I^+(\lambda, r) + (2 + \log(n/p_{\min}))k/n$. The second inequality uses the fact that for $0 \leq u \leq v \leq 1$, if $|u - v| \leq 1/n$ then $|\phi(u) - \phi(v)| \leq (2 + \log n)/n$. This is true, because $\phi'(t) = 1 + \log t$, which implies that

$$
\begin{aligned}
|\phi(v) - \phi(u)| &\leq \int_u^v |(1 + \log t)| \, dt \stackrel{(i)}{\leq} \int_u^v (1 - \log t) dt \stackrel{(ii)}{\leq} \int_0^{1/n} (1 - \log t) dt = \int_0^{1/n} (2 - (1 + \log t)) dt \\
&= \int_0^{1/n} (2 - \phi'(t)) dt = \frac{2}{n} - (\phi(1/n) - \phi(0)) = \frac{2}{n} - \frac{1}{n} \log \left( \frac{1}{n} \right) = \frac{2 + \log n}{n}.
\end{aligned}
$$

The inequality $(i)$ uses the fact that for $t \in [0, 1]$, we have $|1 + \log t| \leq 1 - \log t$, and $(ii)$ uses the fact that the mapping $t \mapsto 1 - \log t$ is decreasing. This completes the proof of the bound for $I^+(\lambda, r)$. The same argument also works for $I^-(\lambda, r)$ and we omit the details. $\qquad \square$

## B. Deferred Proofs

### B.1. Proof of Theorem 3.1

The first part of the result considers the case of playing the Kelly bet at all remaining steps. In particular, let $Z_i^K = h_m(\lambda_m^{\text{Kelly}}, X_i)$ denote the increments of log-wealth under this policy, and observe that $Z_i^K \in [-B, B]$ for all $i$. Now, we note that $\{\tau \leq N\} = \cup_{k=1}^{T} \{\tau \leq t + k\} \supset \{y + \sum_{i=t+1}^{N} Z_i^K \geq b\}$, which implies that

$$
\mathbb{P}(\tau > N) \leq \mathbb{P} \left( \sum_{i=t+1}^{N} (Z_i^K - L_{\max}) \leq \underbrace{(b - y) - TL_{\max}}_{= -\Delta} \right).
$$

Since first part of the result assume that $\Delta \geq B\sqrt{8T \log T}$, an application of Hoeffding's inequality implies

$$
\mathbb{P}(\tau > N) \leq \mathbb{P} \left( \sum_{i=t+1}^{N} Z_i^K - L_{\max} \leq -\Delta \right) \leq \mathbb{P} \left( \sum_{i=t+1}^{N} Z_i^K - L_{\max} \leq -B\sqrt{8T \log T} \right) \leq \frac{1}{T}.
$$

This justifies the first part of Theorem 3.1.

For the second part, let $Z_i = h_m(\lambda_i, X_i)$ with $\mathbb{E}[Z_i \mid \mathcal{F}_{i-1}] = L(\lambda_i)$. Thus, the terms $\{D_i := Z_i - L(\lambda_i) : t + 1 \leq i \leq N\}$ forms a martingale difference sequence. For any $k \geq t + 1$, let $S_k = \sum_{i=t+1}^{k} D_i$ denote the martingale with bounded difference $|D_i| \leq 2B$ for all $i$. Furthermore, by assumption of the second part of the theorem, we know that for any $k \in [T]$, we have

$$
\sum_{i=t+1}^{k} L(\lambda_i) \leq kL_{\max} - (\rho k)\epsilon = k(L_{\max} - \rho\epsilon).
$$

Now, if $\tau \leq N$, for some $k \in [T]$,

$$
b \leq y + \sum_{i=t+1}^{t+k} Z_i = y + S_k + \sum_{i=t+1}^{t+k} L(\lambda_i) \leq y + S_k + k(L_{\max} - \rho\epsilon).
$$

This implies that if $\tau \leq N$, then for some $k \in [T]$,

$$S_k \geq (b - y) - k(L_{\max} - \rho\epsilon) \geq (b - y) - T(L_{\max} - \rho\epsilon) = \rho\epsilon T - \Delta.$$

In other words,

$$\{\tau \leq N\} \subset \left\{ \max_{1 \leq k \leq T} S_k \geq \rho\epsilon T - \Delta \right\} =: \mathcal{E}.$$

To conclude the proof, we claim that the event $\mathcal{E}$ defined above satisfies

$$\mathbb{P}(\tau \leq N) \leq \mathbb{P}(\mathcal{E}) \leq \exp\left( -\frac{(\rho\epsilon T - \Delta)_+^2}{8B^2 T} \right) \leq \frac{1}{2}, \tag{8}$$

where the last inequality follows from the assumption that $\Delta \leq \rho\epsilon T - B\sqrt{8\log 2}$. To prove the upper bound on $\mathbb{P}(\mathcal{E})$, observe that $\{D_i : i \in [T]\}$ form a martingale difference sequence with $|D_j| \leq 2B$ almost surely. Hence, with $S_k = \sum_{i=1}^k D_i$, we know that $\{M_k(\theta) : k \in [T]\}$ with $M_k(\theta) := e^{\theta S_k - B^2\theta^2 k/2}$ for some $\theta > 0$, forms a nonnegative supermartingale sequence with an initial value of 1 by using Hoeffding's Lemma. Hence, an application of Ville's inequality implies that

$$\mathbb{P}\left( \max_{1 \leq k \leq T} M_k(\theta) \geq a \right) \leq \frac{1}{a}, \quad \text{for} \quad a > 0.$$

Now, since $\{\max_{k \in [T]} S_k \geq s\} \subset \{\max_{k \in [T]} M_k(\theta) \geq e^{\theta s - B^2\theta^2 T/2}\}$, the above implies that $\mathbb{P}(\max_{k \in [T]} S_k \geq s) \leq e^{-\theta s + B^2\theta^2 T/2}$. On optimizing over $\theta$, we get the required $e^{-s^2/2B^2 T}$ upper bound used in (8).

### B.2. Proof of Proposition 3.4

To show this result, we will obtain a lower bound on $\mathbb{P}(\tau(\lambda^{\mathrm{agg}}) \leq N)$ and an upper bound on $\mathbb{P}(\tau(\lambda_m^{\mathrm{Kelly}}) \leq N)$, and then show that under the conditions of the proposition, their difference is positive. The assumption of finite $|\mathcal{X}|$ allows us to use elementary type-based arguments for the probabilities of rare events; see (Csiszár, 1998) and (Cover & Thomas, 2006, Chapter 11). Throughout this section, we will use $\mathcal{P}_T(\mathcal{X})$ to denote the set of all types or empirical distributions on $\mathcal{X}$ with denominator $T$, and introduce the term

$$I_T^+(\lambda, r) = \inf_{Q \in \mathcal{Q}_T^+(r, m, \lambda)} D(Q \| P_X), \quad \text{where} \quad \mathcal{Q}_T^+(r, m, \lambda) = \left\{ Q \in \mathcal{P}_T(\mathcal{X}) : \mathbb{E}_Q[h_m(\lambda, X)] \geq r \right\}.$$

We first look at the term $\mathbb{P}(\tau(\lambda^{\mathrm{agg}}) \leq N)$, and observe that

$$\mathbb{P}(\tau(\lambda^{\mathrm{agg}}) \leq N) = \mathbb{P}\left( \bigcup_{k=1}^T \left\{ \sum_{i=t+1}^{t+k} h_m(\lambda^{\mathrm{agg}}, X_i) \geq b - y \right\} \right) \geq \mathbb{P}\left( \frac{1}{T} \sum_{i=t+1}^N h_m(\lambda^{\mathrm{agg}}, X_i) \geq r \right)$$
$$\geq (T+1)^{|\mathcal{X}|} \exp\left( -T I_T^+(\lambda^{\mathrm{agg}}, r) \right). \tag{9}$$

The last inequality above follows from an application of Sanov's theorem for finite alphabets (Cover & Thomas, 2006, Theorem 11.4.1) recalled in Fact A.2.

Similarly, for $\mathbb{P}\left( (\tau(\lambda_m^{\mathrm{Kelly}}) \leq N \right)$, we can use the fact that $y$ is small enough that it is impossible to hit the threshold in fewer than $T/2 = (N - t)/2$ steps as stated in (3). This implies that

$$\mathbb{P}\left( \tau(\lambda_m^{\mathrm{Kelly}}) \leq N \right) = \mathbb{P}\left( \bigcup_{k=T/2+1}^T \left\{ \sum_{i=t+1}^{t+k} h_m(\lambda_m^{\mathrm{Kelly}}, X_i) \geq b - y \right\} \right)$$
$$\leq \sum_{k=T/2+1}^T \mathbb{P}\left( \frac{1}{k} \sum_{i=t+1}^{t+k} h_m(\lambda_m^{\mathrm{Kelly}}, X_i) \geq r\frac{T}{k} \right)$$
$$\leq \sum_{k=T/2+1}^T (k+1)^{|\mathcal{X}|} \exp\left( -k I^+(\lambda_m^{\mathrm{Kelly}}, rT/k) \right), \tag{10}$$

where the first inequality follows from the union bound, and the second inequality again uses Sanov's theorem for finite alphabets. Finally, recalling the definition of $I^+(\cdot, \cdot)$ from (4), observe that for all $k \in \{T/2, \ldots, T\}$,

$$I^+(\lambda_m^{\text{Kelly}}, r) = \inf_{Q \in \mathcal{Q}^+(r, m, \lambda_m^{\text{Kelly}})} D(Q \parallel P_X) \leq \inf_{Q \in \mathcal{Q}^+\left(r\frac{T}{k}, m, \lambda_m^{\text{Kelly}}\right)} D(Q \parallel P_X) = I^+\left(\lambda^{\text{Kelly}}, r\frac{T}{k}\right) = I^+\left(\lambda^{\text{Kelly}}, r\frac{T}{k}\right).$$

Using this along with $(k+1)^{|\mathcal{X}|} \leq (T+1)^{|\mathcal{X}|}$ for all $k \leq T$ in (10), we get

$$\mathbb{P}\left(\tau(\lambda_m^{\text{Kelly}}) \leq N\right) \leq \frac{T}{2}(T+1)^{|\mathcal{X}|} \exp\left(-\frac{T}{2} I^+(\lambda_m^{\text{Kelly}}, r)\right).$$

Combining this with (9), we get

$$\frac{\mathbb{P}\left(\tau(\lambda^{\text{agg}}) \leq N\right)}{\mathbb{P}\left(\tau(\lambda_m^{\text{Kelly}}) \leq N\right)} \geq \frac{(T+1)^{-|\mathcal{X}|} \exp\left(-T I_T^+(\lambda^{\text{agg}}, r)\right)}{\frac{T}{2}(T+1)^{|\mathcal{X}|} \exp\left(-\frac{T}{2} I^+(\lambda_m^{\text{Kelly}}, r)\right)}$$

$$= \exp\left(T\left\{\frac{1}{2} I^+(\lambda_m^{\text{Kelly}}, r) - \frac{\log\left(\frac{T}{2}(T+1)^{2|\mathcal{X}|}\right)}{T} - I_T^+(\lambda^{\text{agg}}, r)\right\}\right).$$

Thus the lower bound on the ratio of the two probabilities is strictly greater than 1, if $I_T^+(\lambda^{\text{agg}}, r) < \frac{1}{2} I^+(\lambda_m^{\text{Kelly}}, r) - \frac{\log\left(\frac{T}{2}(T+1)^{2|\mathcal{X}|}\right)}{T}$. Furthermore, by Lemma A.3, we know that $I_T^+(\lambda^{\text{agg}}, r) \leq I^+(\lambda^{\text{agg}}, r) + C(T, |\mathcal{X}|, p_{\min})$, which implies the required sufficient condition

$$I^+(\lambda^{\text{agg}}, r) \leq \frac{1}{2} I^+(\lambda^{\text{Kelly}}, r) - \frac{\log\left(\frac{T}{2}(T+1)^{2|\mathcal{X}|}\right)}{T} - C(T, |\mathcal{X}|, p_{\min}) =: \frac{1}{2} I^+(\lambda^{\text{Kelly}}, r) - c_T^+,$$

where

$$c_T^+ = \frac{\log\left(\frac{T}{2}(T+1)^{2|\mathcal{X}|}\right)}{T} + \frac{|\mathcal{X}|}{T}\left(2 + \log\left(\frac{T}{\min_{x \in \mathcal{X}} P_X(x)}\right)\right). \tag{11}$$

### B.3. Proof of Proposition 3.6

We follow the same pattern as the proof of Proposition 3.4, and begin by introducing the analog of $I_T^+$ for this case:

$$I_{T/2}^-(\lambda, r) := \inf_{Q \in \mathcal{Q}_{T/2}^-(r, m, \lambda)} D(Q \parallel P_X), \quad \text{where} \quad \mathcal{Q}_{T/2}^-(r, m, \lambda) = \{Q \in \mathcal{P}_{T/2}(\mathcal{X}) : \mathbb{E}_Q[h_m(\lambda, X)] \leq r\},$$

where $\mathcal{P}_{T/2}(\mathcal{X})$ denotes the set of all types or empirical distributions with denominator $T/2$ on $\mathcal{X}$.

For any feasible $\lambda$, let $F(\lambda)$ denote the failure probability $\mathbb{P}(\tau(\lambda) > N)$. First we obtain an upper bound on $F(\lambda^{\text{def}})$.

$$F(\lambda^{\text{def}}) = \mathbb{P}\left(\bigcap_{k=1}^T \left\{\sum_{i=t+1}^{t+k} h_m(\lambda^{\text{def}}, X_i) < b - y\right\}\right) \leq \mathbb{P}\left(\frac{1}{T} \sum_{i=t+1}^N h_m(\lambda^{\text{def}}, X_i) < r\right).$$

An application of Sanov's theorem for finite alphabets implies that

$$F(\lambda^{\text{def}}) \leq (T+1)^{|\mathcal{X}|} \exp\left(-T I^-(\lambda^{\text{def}}, r)\right). \tag{12}$$

Now, let $B_K$ denote the value $\max_{x \in \mathcal{X}} h_m(\lambda_m^{\text{Kelly}}, x)$, and consider the event

$$\mathcal{E} = \left\{\frac{2}{T} \sum_{i=t+1}^{t+T/2} h_m(\lambda_m^{\text{Kelly}}, X_i) \leq r_-\right\}, \quad \text{where} \quad r_- = 2\left(r - \frac{1}{2} B_K\right).$$

Crucially, under this event $\mathcal{E}$, we have

$$y + \sum_{i=t+1}^{t+T/2} h_m(\lambda_m^{\text{Kelly}}, X_i) < y + \frac{T}{2} r_- = y + (b - y) - \frac{T}{2} B_K = b - \frac{T}{2} B_K.$$

In other words, under $\mathcal{E}$, with $T/2$ steps remaining, the Kelly betting scheme is at least $(T/2)B_K$ away from the threshold, and it is thus impossible for this scheme to hit the threshold. Thus,

$$F(\lambda_m^{\text{Kelly}}) \geq \mathbb{P}(\mathcal{E}) \geq (T/2 + 1)^{-|\mathcal{X}|} \exp\left(-\frac{T}{2} I_{T/2}^-(\lambda^{\text{Kelly}}, r_-)\right),$$

by another application of Sanov's Theorem for finite alphabets. Combining this with (12), we get

$$
\begin{aligned}
\frac{F(\lambda_m^{\text{Kelly}})}{F(\lambda^{\text{def}})} &\geq \frac{(T/2) + 1)^{-|\mathcal{X}|} \exp\left(-\frac{T}{2} I_{T/2}^-(\lambda^{\text{Kelly}}, r_-)\right)}{(T+1)^{|\mathcal{X}|} \exp\left(-T I^-(\lambda^{\text{def}}, r)\right)} \\
&\geq \exp\left(T\left(I^-(\lambda^{\text{def}}, r) - \frac{1}{2} I_{T/2}^-(\lambda_m^{\text{Kelly}}, r_-) - \frac{\log\left(\frac{T}{2}(T+1)^{2|\mathcal{X}|}\right)}{T}\right)\right).
\end{aligned}
$$

Thus, if there exists a $\lambda^{\text{def}}$ satisfying the condition $I^-(\lambda^{\text{def}}, r) - \frac{1}{2} I_{T/2}^-(\lambda_m^{\text{Kelly}}, r_-) - \frac{\log\left(\frac{T}{2}(T+1)^{2|\mathcal{X}|}\right)}{T} > 0$, then the failure probability of Kelly bet, $F(\lambda_m^{\text{Kelly}})$, is exponentially larger than the failure probability of the defensive bet, $F(\lambda^{\text{def}})$. To complete the proof, we again use Lemma A.3 to obtain

$$I_{T/2}^-(\lambda_m^{\text{Kelly}}, r_-) \leq I^-(\lambda_m^{\text{Kelly}}, r_-) + \frac{2|\mathcal{X}|}{T}\left(2 + \log\left(\frac{T}{2 \min_{x \in \mathcal{X}} P_X(x)}\right)\right).$$

Thus, we obtain the following sufficient condition for the failure probability of $\lambda^{\text{def}}$ to be exponentially smaller than Kelly bet: $I^-(\lambda^{\text{def}}, r) > \frac{1}{2} I^-(\lambda_m^{\text{Kelly}}, r_-) + c_T^-$, where

$$c_T^- := \frac{\log\left(\frac{T}{2}(T+1)^{2|\mathcal{X}|}\right)}{T} + \frac{|\mathcal{X}|}{T}\left(2 + \log\left(\frac{T}{2 \min_{x \in \mathcal{X}} P_X(x)}\right)\right). \tag{13}$$

This completes the proof.

## C. Hedging over $\epsilon$-schedules.

Let the schedule class be indexed by

$$\mathcal{K} = \{(\eta, q) : \eta \in \{0.25, 0.50, 0.75\}, \ q \in \{1, 2\}\}, \qquad |\mathcal{K}| = K = 6,$$

where $q = 1$ corresponds to linear growth and $q = 2$ to quadratic growth, and $\eta$ is the aggressiveness onset fraction. For each $k = (\eta, q) \in \mathcal{K}$ we define

$$\epsilon_t^{(k)} = \begin{cases} 0, & t < \eta N, \\ \left(\frac{t - \eta N}{(1 - \eta)N}\right)^q, & t \in [\eta N, N], \end{cases}$$

so that $\epsilon_{\eta N}^{(k)} = 0$ and $\epsilon_N^{(k)} = 1$.

Fix $m \in [0, 1]$. For each schedule $k \in \mathcal{K}$ we run a separate mixed-strike betting strategy producing a predictable bet sequence $\{\lambda_t^{(k)}(m)\}_{t \geq 1} \subset \Lambda_m$. Concretely, letting $\widehat{\lambda}_t(m)$ denote the empirical Kelly estimate and $\lambda_{\max} \in \Lambda_m$ an aggressive bet, we implement the $\epsilon$-greedy rule by drawing (independently of the data)

$$U_t^{(k)} \sim \text{Bernoulli}(\epsilon_t^{(k)}), \qquad \lambda_t^{(k)}(m) = \left(1 - U_t^{(k)}\right) \widehat{\lambda}_t(m) + U_t^{(k)} \lambda_{\max}.$$

Equivalently, one may view $\lambda_t^{(k)}(m)$ as predictable with respect to an enlarged filtration that includes the auxiliary randomness $\{U_t^{(k)}\}$.) Each schedule induces a wealth process

$$W_0^{(k)}(m) = 1, \qquad W_t^{(k)}(m) = W_{t-1}^{(k)}(m)\Big(1 + \lambda_t^{(k)}(m)\big(X_t - m\big)\Big), \quad t \geq 1.$$

We hedge by splitting the initial unit capital uniformly across schedules and tracking the total (mixture) wealth

$$W_t(m) \;=\; \frac{1}{K}\sum_{k\in\mathcal{K}} W_t^{(k)}(m).$$

Under the null $H_0 : \mathbb{E}[X] = m$, each $\{W_t^{(k)}(m)\}_{t\geq 0}$ is a nonnegative test martingale and therefore their convex mixture $\{W_t(m)\}_{t\geq 0}$ is also a nonnegative test martingale. Hence, by Ville's inequality,

$$\mathbb{P}_{H_0}\left(\sup_{t\leq N} W_t(m) \geq \frac{1}{\alpha}\right) \leq \alpha.$$

We thus use the hedged stopping rule

$$\tau_m \;=\; \inf\left\{t \in \{1,\ldots,N\} : \log W_t(m) \geq \log\left(\frac{1}{\alpha}\right)\right\},$$

and reject $H_{0,m}$ when $\tau_m \leq N$.

Finally, the hedged wealth is within an additive $\log K$ of the best schedule in log-wealth:

$$\log W_t(m) = \log\left(\frac{1}{K}\sum_k W_t^{(k)}(m)\right) \;\geq\; \max_k \log W_t^{(k)}(m) - \log K,$$

so the price of hedging over $K = 6$ schedules is small (an additive $\log 6$ in log-evidence) while providing robustness to the unknown $P_X$ and problem difficulty.

## D. Algorithmic Implementations

We implement the *Linear-$\epsilon$* baseline using a single mixed-strike schedule. Fixing an onset fraction $\eta = 0.5$, we define

$$
\epsilon_t^{\text{lin}} = \begin{cases} 0, & t < \eta N, \\ \dfrac{t - \eta N}{(1 - \eta)N}, & t \in [\eta N,\, N], \end{cases}
$$

so $\epsilon_{\eta N}^{\text{lin}} = 0$ and $\epsilon_N^{\text{lin}} = 1$. At each time $t$, we draw an auxiliary coin $U_t \sim \text{Bernoulli}(\epsilon_t^{\text{lin}})$ independently of the data, and set

$$
\lambda_t(m) = (1 - U_t)\,\widehat{\lambda}_t(m) + U_t\,\lambda_{\max},
$$

where $\widehat{\lambda}_t(m)$ is the predictable empirical-Kelly estimate and $\lambda_{\max} \in \Lambda_m$ is an aggressive (endpoint/all-in) bet. We clip $\lambda_t(m)$ to $\Lambda_m$ (with the same $\varepsilon$-margin used elsewhere) to ensure $1 + \lambda_t(X_t - m) \geq \varepsilon$ for all $X_t \in [0, 1]$. We update wealth via $W_t = W_{t-1}\big(1 + \lambda_t(m)(X_t - m)\big)$ and reject when $\max_{t \leq N} W_t \geq 1/\alpha$.

To reduce sensitivity to the choice of $(\eta, \text{trend})$, we hedge across a finite family of schedules $\mathcal{K}$ (e.g., $\eta \in \{0.25, 0.50, 0.75\}$ and trend $q \in \{1, 2\}$, so $K = |\mathcal{K}| = 6$). For each $k \in \mathcal{K}$, we run an independent mixed-strike strategy with its own coin flips $U_t^{(k)} \sim \text{Bernoulli}(\epsilon_t^{(k)})$ and bets

$$
\lambda_t^{(k)}(m) = (1 - U_t^{(k)})\,\widehat{\lambda}_t(m) + U_t^{(k)}\,\lambda_{\max}, \qquad W_t^{(k)}(m) = W_{t-1}^{(k)}(m)\Big(1 + \lambda_t^{(k)}(m)(X_t - m)\Big).
$$

We then form the *hedged* (mixture) wealth by splitting unit capital uniformly:

$$
W_t(m) = \frac{1}{K} \sum_{k \in \mathcal{K}} W_t^{(k)}(m).
$$

Under $H_0 : \mathbb{E}[X] = m$, each $\{W_t^{(k)}(m)\}$ is a test martingale, hence their convex mixture $\{W_t(m)\}$ is also a test martingale and remains anytime-valid. We reject when $\max_{t \leq N} W_t(m) \geq 1/\alpha$.

Also, the STaR procedure (Voráček & Orabona, 2025) is designed for *one-sided* alternatives and outputs a one-sided predictable bet magnitude $\tilde{\lambda}_t(m) \geq 0$, clipped to the safe range. To use STaR as a baseline in our *two-sided* setting $H_1 : \mu \neq m$, we keep the same rule for the magnitude and choose the direction using the sign of the current mean estimate:

$$
s_t(m) = \text{sign}\big(\widehat{\mu}_{t-1} - m\big), \qquad \lambda_t^{\text{STaR-2s}}(m) = \text{clip}\big(s_t(m)\,\tilde{\lambda}_t(m),\, \Lambda_m\big),
$$

with $\text{sign}(0) = 0$. This yields a two-sided version of STaR while preserving predictability and the constraint $\lambda_t(m) \in \Lambda_m$.

## E. Experimental Details

Now, we describe the implementation details for the universal DQN betting policy introduced in Section 4.3.

### E.1. Environment.

We treat each horizon-aware test run as an episodic MDP of length at most $N$. At time $t \in \{0, 1, \ldots, N-1\}$ we choose a predictable bet $\lambda_{t+1}(m) \in \Lambda_m$ and update wealth via

$$
W_{t+1}(m) = W_t(m)\big(1 + \lambda_{t+1}(m)(X_{t+1} - m)\big), \qquad Y_t := \log W_t(m),
$$

with $W_0 = 1$ (so $Y_0 = 0$) and threshold $\log(1/\alpha)$. An episode terminates upon first threshold crossing, $\tau = \inf\{t \geq 1 : Y_t \geq \log(\frac{1}{\alpha})\}$, or at the deadline $N$. We use a sparse terminal reward

$$
R_t = \mathbf{1}\{Y_t \geq \log(\frac{1}{\alpha})\},
$$

issued at the first hitting time (and $0$ otherwise). Since the episode stops at the first hit, the undiscounted return (we set $\gamma = 1$) equals the indicator of rejection by the deadline, so maximizing expected return is equivalent to maximizing $\mathbb{P}(\tau \leq N)$.

### E.2. Actions.

The DQN outputs a discrete action $a_t \in \{0, 1, 2\}$ which is mapped to a feasible bet $\lambda_t(m) \in \Lambda_m$ using the current predictable empirical Kelly estimate $\widehat{\lambda}_t(m)$ and an all-in bet $\lambda_{\mathrm{end},t}(m)$:

$$a_t = 0: \ \lambda_t = \tfrac{1}{2}\widehat{\lambda}_t(m), \qquad a_t = 1: \ \lambda_t = \widehat{\lambda}_t(m), \qquad a_t = 2: \ \lambda_t = \lambda_{\mathrm{end},t}(m).$$

The endpoint bet is directional, chosen by the sign of $\widehat{\mu}_{t-1} - m$ and clipped to the safe range $\Lambda_m = [-(1 - \varepsilon)/(1 - m), (1 - \varepsilon)/m]$ (we use $\varepsilon = 10^{-3}$) so that $1 + \lambda_t(X_t - m) \geq \varepsilon > 0$ for all $X_t \in [0, 1]$, preventing numerical issues from $\log(0)$ and preserving anytime-validity by construction.

At each time step we compute $\widehat{\mu}_{t-1}$ and a stable Taylor/Kelly-style estimate using only past data. Let $S_{t-1} = \sum_{i=1}^{t-1}(X_i - m)$ and $V_{t-1} = \sum_{i=1}^{t-1}(X_i - m)^2$. We use

$$\widehat{\lambda}_t(m) = \mathrm{clip}\left(\frac{S_{t-1}}{V_{t-1}}, \ \Lambda_m\right),$$

with the convention $\widehat{\lambda}_t(m) = 0$ when $V_{t-1} = 0$.

### E.3. State.

Although the oracle DP would use the state $(t, Y_t)$ when $(m, P_X)$ is fixed, the data-generating distribution $P_X$ is unknown in practice. Instead, we feed the DQN a predictable feature vector $\phi_t \in \mathbb{R}^{22}$, computed only from $\mathcal{F}_{t-1}$ and the current log-wealth $Y_{t-1}$. The features summarize: (i) the mean gap $\widehat{\mu}_{t-1} - m$ (and its magnitude), (ii) how far we are from the rejection threshold, including a normalized distance-to-threshold and the per-step log-wealth increase needed to reach it by time $N$, (iii) the remaining time fraction $(N - 1 - t)/(N - 1)$, (iv) variance proxies around $m$ and an SNR-style normalized gap, (v) the current empirical Kelly and endpoint bets, $\widehat{\lambda}_t(m)$ and $\lambda_{\mathrm{end},t}(m)$ along with their difference, (vi) simple Taylor-style proxies for the one-step growth advantage of endpoint versus Kelly and (vii) additional shape summaries from empirically centered moments (skewness, excess kurtosis), and a Beta concentration proxy $\log \widehat{\kappa}$. Because every component is computed predictably (with no dependence on $X_t$), the learned policy remains compatible with anytime-valid inference.

### E.4. Synthetic Training Distribution.

Training is performed purely on synthetic episodes. We consider two data-generating families:

- Beta world: $X_i \sim \mathrm{Beta}(\mathrm{conc} \cdot \mu, \ \mathrm{conc} \cdot (1 - \mu))$.

- Beta-mixture world: each $X_i$ is drawn from a $50/50$ mixture of $\mathrm{Beta}(\mathrm{conc} \cdot \mu, \mathrm{conc} \cdot (1 - \mu))$ and $\mathrm{Beta}(2\mathrm{conc} \cdot \mu, 2\mathrm{conc} \cdot (1 - \mu))$.

Each rollout batch uses Beta or Beta-mixture with probability $1/2$ each.

### E.5. Training

To train a single policy that generalizes across horizons and nulls, we sample rollout configurations. For each rollout batch we sample $(N, m, \mu)$ as follows:

1. Sample $N$ log-uniformly from a range $[N_{\min}, N_{\max}]$

2. Sample $m$ uniformly from $[m_{\min}, m_{\max}]$ (away from the boundaries)

3. Sample a difficulty multiplier $c \sim \mathrm{Unif}[c_{\min}, c_{\max}]$ and set

$$|\mu - m| \approx \sqrt{\frac{2\,\sigma_{\mathrm{proxy}}^2\, c\, \log(1/\alpha)}{N}},$$

with a random sign, where $\sigma_{\mathrm{proxy}}^2$ is a variance proxy under the sampled world then clip $\mu$ into $[\mu_{\min}, \mu_{\max}]$.

This coupling keeps problems across different $N$ at comparable difficulty scales. We also randomize the concentration parameter conc within a specified range.

We implement DQN in PyTorch with a 2-hidden-layer MLP:

$$22 \xrightarrow{\text{ReLU}} 256 \xrightarrow{\text{ReLU}} 128 \rightarrow |\mathcal{A}|,$$

where $|\mathcal{A}| = 3$ actions. We use Double DQN setting (Van Hasselt et al., 2016), with Huber loss (Huber, 1992), and the AdamW optimizer (Loshchilov & Hutter, 2019). We use a replay buffer and update the target network every fixed number of gradient steps. We also clip gradient norms to stabilize training.

Exploration is $\epsilon$-greedy over actions with an exponential decay schedule: $\epsilon_k = \max(\epsilon_{\min}, \epsilon_0 \cdot \rho^k)$ as a function of episode index $k$. We collect experience using vectorized rollouts with multiple environments in parallel and train off-policy from the replay buffer. For model selection, we evaluate checkpoints using the *greedy* policy ($\epsilon = 0$) on a fixed evaluation set (fixed random seed), and select the checkpoint with the highest estimated hit-rate $\mathbb{P}(\tau \leq N)$. We also adapt the learning rate using a plateau scheduler based on this evaluation metric.

Training takes about three hours on a single L40s GPU. Although the MLP is small enough to train on CPU, we use a GPU for faster training.

*Table 1.* DQN training hyperparameters used in our main runs.

| Hyperparameter | Setting |
|---|---|
| Actions | $\{\widehat{\lambda}/2,\ \widehat{\lambda},\ \lambda_{\text{end}}\}$ |
| Feature dimension | 22 |
| Network | MLP (256, 128) with ReLU activation |
| Discount | $\gamma = 1$ |
| Loss | Huber Loss |
| Optimizer | AdamW, lr $= 3 \cdot 10^{-4}$ |
| Replay buffer capacity | $3.2 \times 10^6$ |
| Min. buffer before training | $4 \times 10^4$ |
| Batch size | 512 |
| Target updates | Hard update every 1000 train steps |
| Exploration | $\epsilon_0 = 1.0$, $\epsilon_{\min} = 0.02$, decay 0.99998 / episode |
| Training episodes | 550,000 |
| Eval for checkpointing | 4000 greedy episodes |
| $N$ range | Log-uniform in $[100, 350]$ |
| $m$ range | Uniform in $[0.01, 0.99]$ |
| Difficulty multiplier $c$ | Uniform in $[0.70, 1.30]$ |
| $\mu$ Clipping | $[0.01, 0.99]$ |
| conc range | in $[0.1, 11.0]$ |
| World | Beta vs. Beta-mixture (50/50 per rollout batch) |

# F. Additional Results

## F.1. Alternative Reward Functions

In Figure 6, we plot the rejection probability, $P(\text{reject by time } t)$, for the original DQN and two additional DQN variants trained with modified rewards: DQN (binary reward, $r = 1$ if the threshold is crossed by the deadline), DQN-EB (early-bonus reward, $r = 1 + (1 - t/N)$), and DQN-U (urgency reward, $r = 1 - t/N$), together with the existing baselines. The distributional setting is the same as in Figure 3: $m = 0.45$, $\mu = 0.40$, $N = 100$, $\alpha = 0.05$, in the Beta-mixture world.

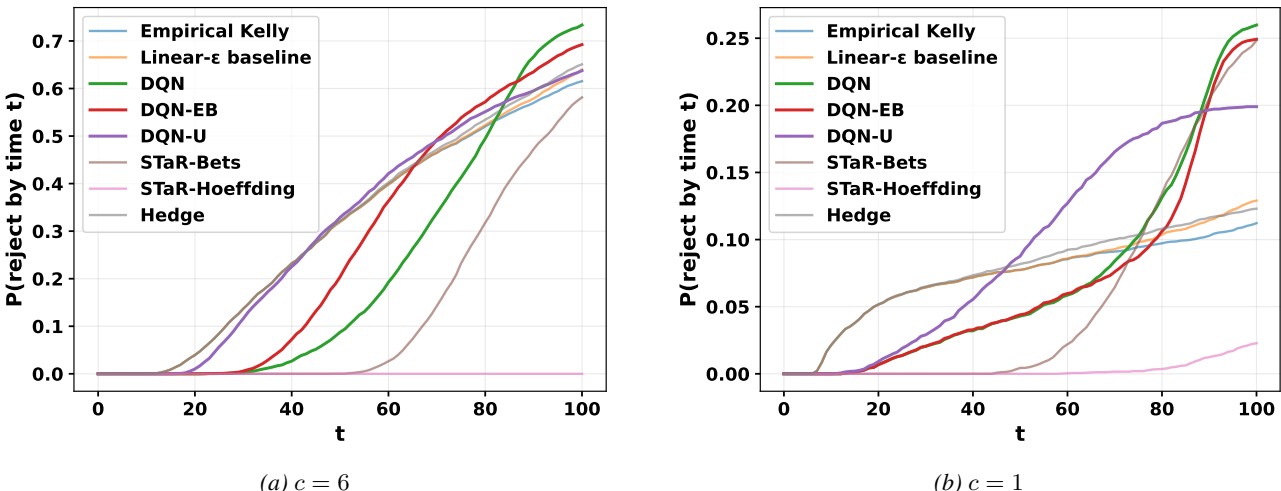

*(a) c = 6*                                 *(b) c = 1*

*Figure 6.* Reward-shaping comparison in the Beta-mixture world. Curves show the cumulative rejection probability $P(\tau \leq t)$, where $\tau$ is the first threshold-crossing time. Both panels use $m = 0.45$, $\mu_X = 0.40$, $N = 100$, and $\alpha = 0.05$; only the concentration $c$ varies. We compare the original DQN, trained with binary deadline reward $\mathbf{1}\{\tau \leq N\}$, with two reward-shaped variants: DQN-EB, with early-bonus reward $\mathbf{1}\{\tau \leq N\}[1 + (1 - \tau/N)]$, and DQN-U, with urgency reward $\mathbf{1}\{\tau \leq N\}(1 - \tau/N)$, along with the existing baselines.

## F.2. Generalization to Other Distributions

We evaluate the DQN policy on out-of-distribution samples drawn from a logit-normal distribution in Figure 7. A random variable $P \in (0, 1)$ is said to follow a logit-normal distribution with parameters $(\mu_X, \sigma_X)$ if $\text{logit}(P) = \log\left(\frac{P}{1-P}\right) \sim \mathcal{N}(\mu_X, \sigma_X^2)$. Equivalently, if $X \sim \mathcal{N}(\mu_X, \sigma_X^2)$, then $P = \frac{e^X}{1+e^X}$ has a logit-normal distribution. The policy was trained on the Beta and Beta-mixture families, and is evaluated here on the logit-normal distribution.

Furthermore, to test whether the approach extends beyond Beta and Beta-mixture settings to discrete distributions, we universally train a DQN policy on Bernoulli families over all $\mu_X, m \in [0, 1]$. We then evaluate it on Bernoulli distributions with $\mu_X \in \{0.10, 0.30, 0.50\}$ at $\alpha = 0.05$. As in the Beta-family experiments, the learned DQN policy also yields power improvements in the Bernoulli setting as seen in Figure 8.

## F.3. Real Data Experiments

**Data domain 1: DNA methylation** (NCBI GEO: GSE33896; Figures 9–11). We use genome-wide CpG methylation beta values. Each observation is a continuous beta value in $[0, 1]$, where $0$ indicates a fully unmethylated CpG site and $1$ indicates a fully methylated site. We test three datasets spanning adipose-derived stem cells, induced osteocytes, and a rhabdomyosarcoma cell line. These empirical distributions are strongly heterogeneous, with pronounced skewness and bimodality, and arise from a completely different data-generating process than the synthetic families used in training. We provide histograms for the empirical distributions along with the power results.

**Data domain 2: Daily relative humidity** (NOAA U.S. Climate Reference Network; Figures 12–14). We use daily average relative humidity values from three stations spanning distinct climates: desert (AZ Yuma), mountain (CO Boulder), and temperate Northeast (NY Millbrook). We scale the raw measurements from $[0\%, 100\%]$ to $[0, 1]$ by dividing by 100. These yield right-skewed, near-symmetric, and mildly left-skewed distributions, respectively. Because our framework assumes i.i.d. observations, we evaluate these data by resampling individual daily measurements with replacement from the observed

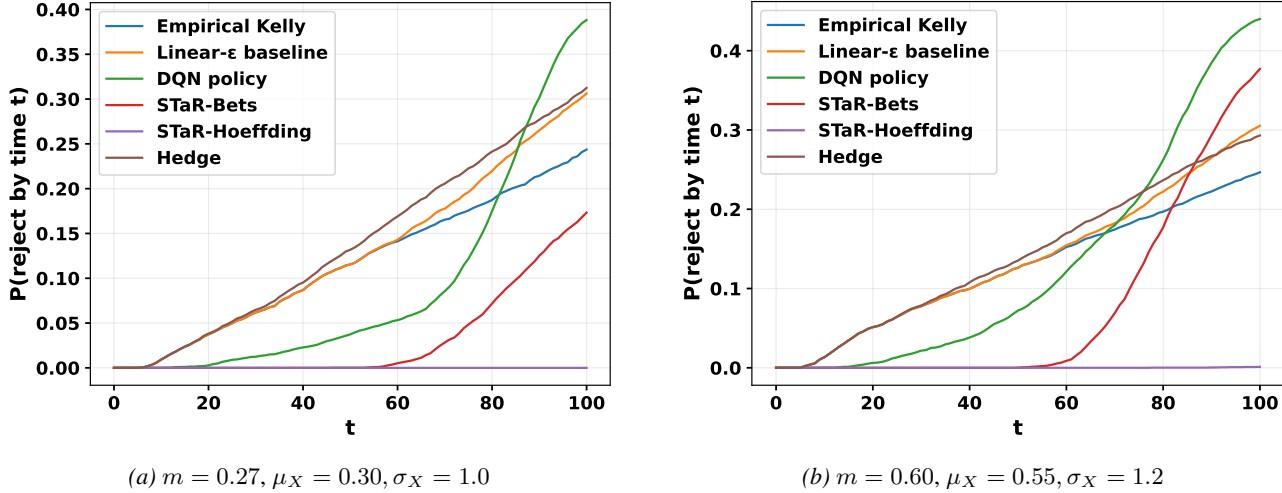

*(a) $m = 0.27, \mu_X = 0.30, \sigma_X = 1.0$*  *(b) $m = 0.60, \mu_X = 0.55, \sigma_X = 1.2$*

*Figure 7.* We evaluate the DQN policy on out-of-distribution samples drawn from a logit-normal distribution. A random variable $P \in (0, 1)$ is said to follow a logit-normal distribution with parameters $(\mu_X, \sigma_X)$ if $\text{logit}(P) = \log\left(\frac{P}{1-P}\right) \sim \mathcal{N}(\mu_X, \sigma_X^2)$. Equivalently, if $X \sim \mathcal{N}(\mu_X, \sigma_X^2)$, then $P = \frac{e^X}{1+e^X}$ has a logit-normal distribution. The policy was trained on the Beta and Beta-mixture families, and is evaluated here on the logit-normal distribution. For **(a)**, $\mu_X = 0.30$ and $\sigma_X = 1.0$; for **(b)**, $\mu_X = 0.55$ and $\sigma_X = 1.2$. In both cases, DQN achieves higher power, even though the evaluation distribution is out of distribution relative to the training distributions.

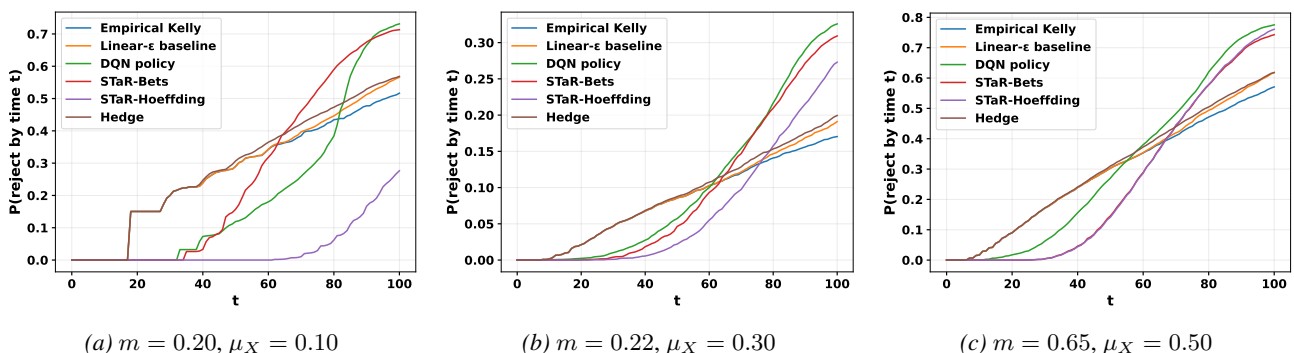

*(a) $m = 0.20, \mu_X = 0.10$*  *(b) $m = 0.22, \mu_X = 0.30$*  *(c) $m = 0.65, \mu_X = 0.50$*

*Figure 8.* Instead of focusing on Beta/Beta-mixture distributions, we universally train a DQN policy on Bernoulli distribution families (for all $\mu_X, m \in [0, 1]$). We evaluate for Bernoulli distributions with $\mu_X = 0.10, \mu_X = 0.30$ and $\mu_X = 0.50$ with $\alpha = 0.05$. Just like for Beta families, DQN learned policy yields power improvements in the Bernoulli case too.

record. This isolates transfer to the real marginal distribution rather than robustness to temporal dependence, which is outside the current scope of this work.

### F.4. Extended Action Space

Our choice of the 3-action set is due to the fact that it is the smallest discrete set that directly matches the three theory-motivated regimes in the phase diagram: conservative, approximately Kelly, and aggressive betting. To investigate the question of whether larger action set yield better policy, we trained an expanded 9-action version and compared it with our original 3-action policy in Figure 15. Empirically, the larger action space achieved power improvements similar to those of the 3-action policy relative to existing baselines, but it behaved somewhat differently: the 9-action version attained a higher rejection probability at earlier times, but a slightly lower rejection probability at the deadline.

Our interpretation is that, under our current synthetic training setup, the benefit of expanding the action space may be limited if the total training time ( 3hours) and synthetic dataset size are held constant. Intuitively, in this sparse-reward RL setting, a larger action space makes optimization harder and increases both data and training requirements. Thus, with fixed compute, data and a simple DQN architecture, the larger-action policy may simply be less well optimized.

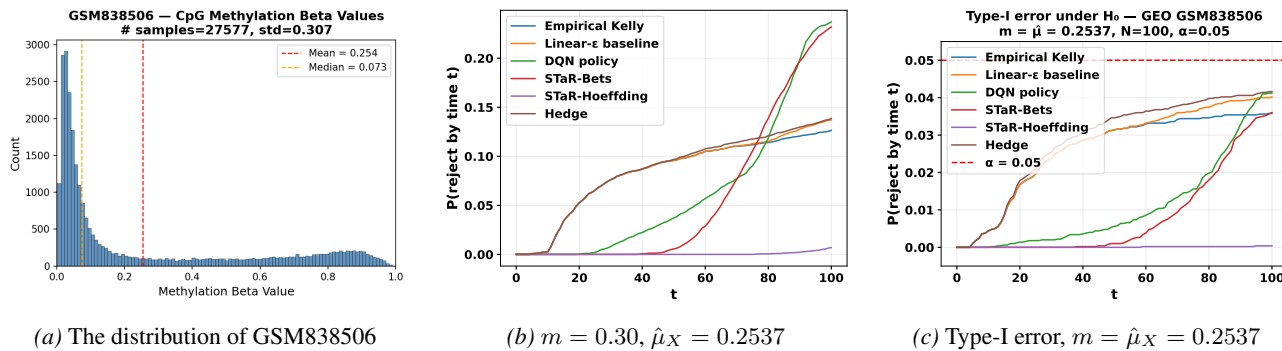

*(a)* The distribution of GSM838506     *(b)* $m = 0.30$, $\hat{\mu}_X = 0.2537$     *(c)* Type-I error, $m = \hat{\mu}_X = 0.2537$

*Figure 9.* Real-data evaluation on CpG methylation beta values from GEO sample GSM838506 (adipose-derived stem cells, donor 1; GSE33896, Illumina HumanMethylation27 BeadChip). (a) The empirical distribution is heavily right-skewed with a bimodal structure. (b) Rejection curves for $H_0: \mu = 0.30$ with randomly $N = 100$ subsampled observations and $\alpha = 0.05$. (c) Type-I error curves for $H_0: \mu = \hat{\mu}_X$ with $N = 100$ and $\alpha = 0.05$. We conducted 5,000 trials to create both rejection and Type-I error curves.

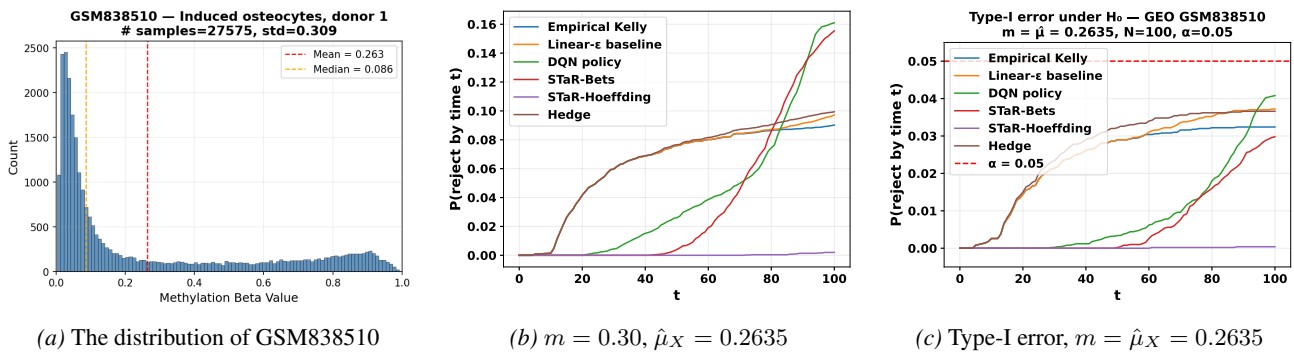

*(a)* The distribution of GSM838510     *(b)* $m = 0.30$, $\hat{\mu}_X = 0.2635$     *(c)* Type-I error, $m = \hat{\mu}_X = 0.2635$

*Figure 10.* Real-data evaluation on CpG methylation beta values from GEO sample GSM838510 (in vitro induced osteocytes, donor 1; GSE33896). (a) The distribution shape is similar to the undifferentiated stem cells but with a slightly higher mean. (b) Rejection curves for $H_0: \mu = 0.30$ with $N = 100$ and $\alpha = 0.05$. (c) Type-I error curves for $H_0: \mu = \hat{\mu}_X$ with $N = 100$ and $\alpha = 0.05$. We conducted 5,000 trials to create both rejection and Type-I error curves.

### F.5. DQN on Longer Deadlines

We evaluate the same DQN policy used in the main experiments at longer deadlines, $N \in \{250, 300, 350\}$, as shown in Figure 16. The data-generating process is the same Beta-mixture setting used in Figure 3. The results show that DQN continues to achieve improved power under these extended horizons.

### F.6. DQN on Different Significance Levels

We also demonstrate that DQN can be adapted to different significance levels. In this experiment (Figure 17), we set $\alpha = 0.01$ while using the same Beta-mixture data-generating setting as in Figure 3. The results show that DQN continues to improve power at this tighter level.

### F.7. Type-I Error of DQN

Figure 18 provides explicit null-calibration plots of $\mathbb{P}(\text{reject by } t)$ under $H_0$ for several null values. Across all settings, the Type-I error remains below $\alpha = 0.05$, as guaranteed by the anytime-validity of the methods. At the same time, DQN approaches the threshold more closely than competing methods, suggesting that it uses the allowable Type-I error budget more effectively, which helps explain its stronger power.

### F.8. DQN Training

We now present the training dynamics. Note that we never reduce the exploration rate to zero. Instead, we anneal it only down to $0.02$. Therefore, the validation hit rate, computed with an exploration rate of $0$, is higher than the training hit rate.

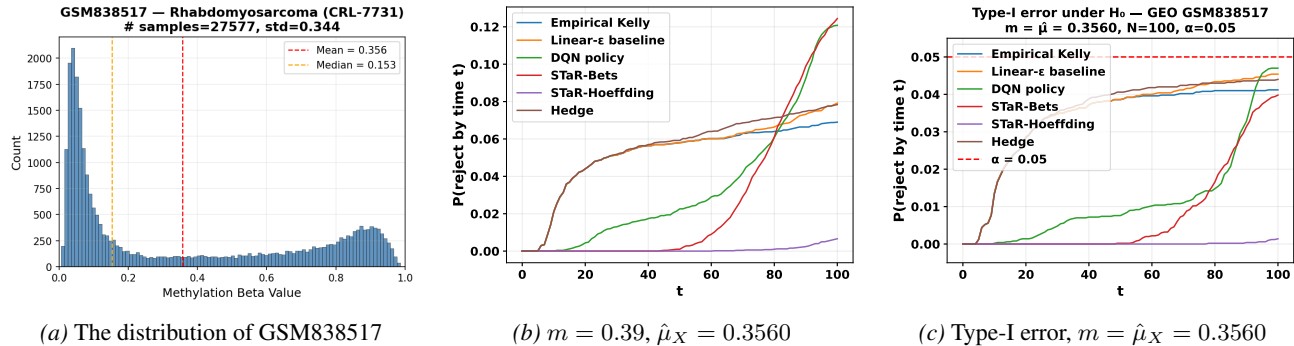

*(a)* The distribution of GSM838517      *(b)* $m = 0.39$, $\hat{\mu}_X = 0.3560$      *(c)* Type-I error, $m = \hat{\mu}_X = 0.3560$

*Figure 11.* Real-data evaluation on CpG methylation beta values from GEO sample GSM838517 (rhabdomyosarcoma cell line CRL-7731; GSE33896). (a) The cancer cell line exhibits a more pronounced bimodal distribution with higher mean and the largest standard deviation among all samples. (b) Rejection curves for $H_0 : \mu = 0.39$ with $N = 100$ and $\alpha = 0.05$. (c) Type-I error curves for $H_0 : \mu = \hat{\mu}_X$ with $N = 100$ and $\alpha = 0.05$. We conducted 5,000 trials to create both rejection and Type-I error curves.

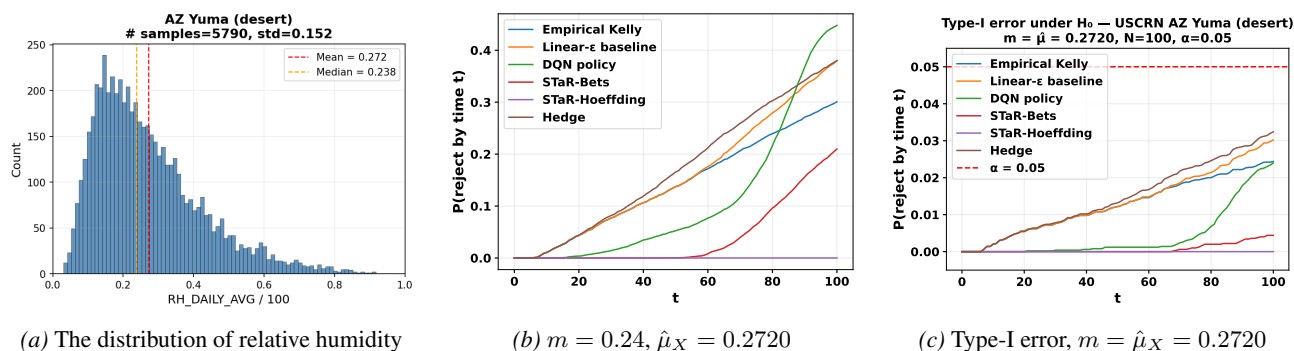

*(a)* The distribution of relative humidity      *(b)* $m = 0.24$, $\hat{\mu}_X = 0.2720$      *(c)* Type-I error, $m = \hat{\mu}_X = 0.2720$

*Figure 12.* Real-data evaluation on daily average relative humidity from the USCRN station at Yuma, AZ (desert climate, 2007–2025). (a) The distribution is right-skewed with low mean. (b) Rejection curves for $H_0 : \mu = 0.24$ with $N = 100$ and $\alpha = 0.05$. (c) Type-I error curves for $H_0 : \mu = \hat{\mu}_X$ with $N = 100$ and $\alpha = 0.05$. We conducted 5,000 trials to create both rejection and Type-I error curves.

We chose the highest return yielding validation checkpoint as the final checkpoint.

### F.9. Qualitative DQN Actions

We also visualize the DQN actions in Figures 20–22. We see that DQN policy is usually more conservative early on, with more aggresive actions in later stages.

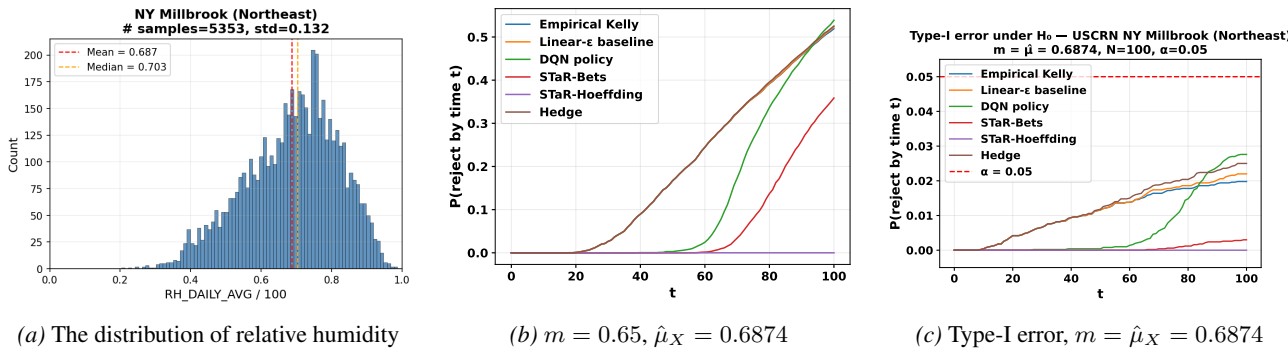

*(a)* The distribution of relative humidity     *(b)* $m = 0.65$, $\hat{\mu}_X = 0.6874$     *(c)* Type-I error, $m = \hat{\mu}_X = 0.6874$

*Figure 13.* Real-data evaluation on daily average relative humidity from the USCRN station at Millbrook, NY (temperate Northeast climate, 2004–2025). (a) The distribution is approximately symmetric with a mild left skew. (b) Rejection curves for $H_0 \colon \mu = 0.65$ with $N = 100$ and $\alpha = 0.05$. (c) Type-I error curves for $H_0 \colon \mu = \hat{\mu}_X$ with $N = 100$ and $\alpha = 0.05$. We conducted 5,000 trials to create both rejection and Type-I error curves.

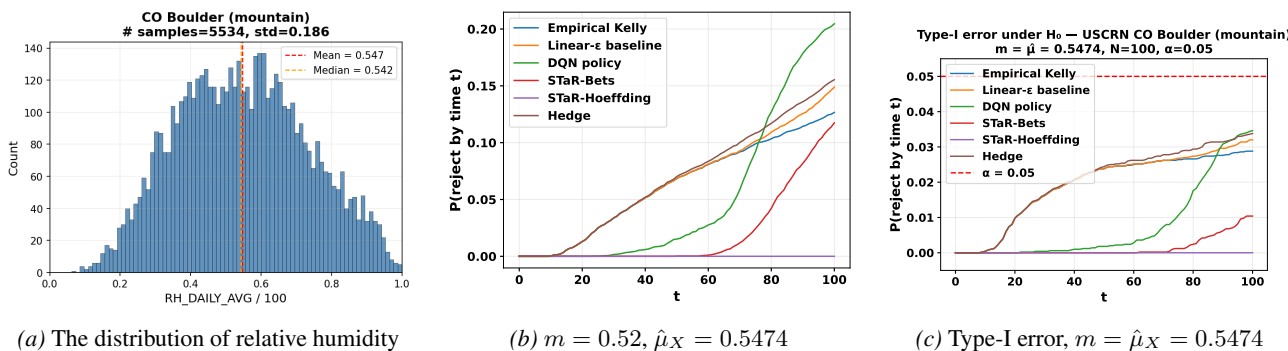

*(a)* The distribution of relative humidity     *(b)* $m = 0.52$, $\hat{\mu}_X = 0.5474$     *(c)* Type-I error, $m = \hat{\mu}_X = 0.5474$

*Figure 14.* Real-data evaluation on daily average relative humidity from the USCRN station at Boulder, CO (mountain climate, 2003–2025). (a) The distribution is nearly symmetric with the highest variance among all USCRN stations tested. (b) Rejection curves for $H_0 \colon \mu = 0.52$ with $N = 100$ and $\alpha = 0.05$. (c) Type-I error curves for $H_0 \colon \mu = \hat{\mu}_X$ with $N = 100$ and $\alpha = 0.05$. We conducted 5,000 trials to create both rejection and Type-I error curves.

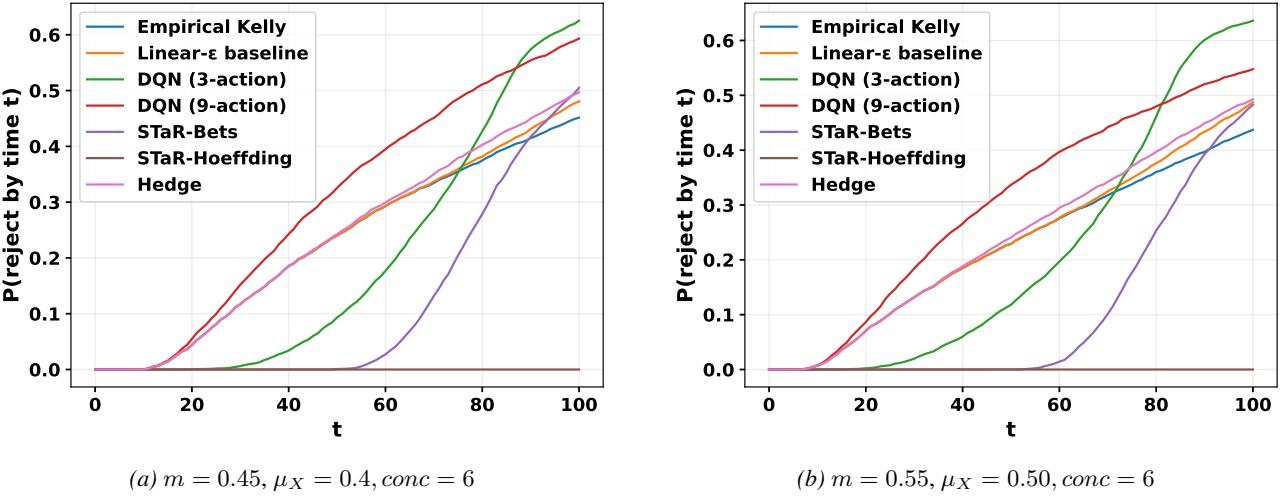

*(a)* $m = 0.45$, $\mu_X = 0.4$, $conc = 6$        *(b)* $m = 0.55$, $\mu_X = 0.50$, $conc = 6$

*Figure 15.* We compare two DQN policies: one trained with the original 3-action space, consisting of $\widehat{\lambda}_t(m)/2$, $\widehat{\lambda}t(m)$, and $\lambda\mathrm{max}$, and another with an expanded 9-action space, namely $\widehat{\lambda}_t(m)/8$, $\widehat{\lambda}_t(m)/4$, $\widehat{\lambda}_t(m)/2$, $\widehat{\lambda}_t(m)$, $\widehat{\lambda}_t(m)$, $\widehat{\lambda}_t(m) \times \frac{5}{4}$, $\widehat{\lambda}_t(m) \times \frac{6}{4}$, $\widehat{\lambda}t(m) \times \frac{7}{4}$, and $\lambda\mathrm{max}$. The distribution in the plot is Beta and $\alpha = 0.05$. We observe that the larger action space yields similar performance but does not lead to an overall improvement. Therefore, for simplicity, we use the 3-action baseline in the paper.

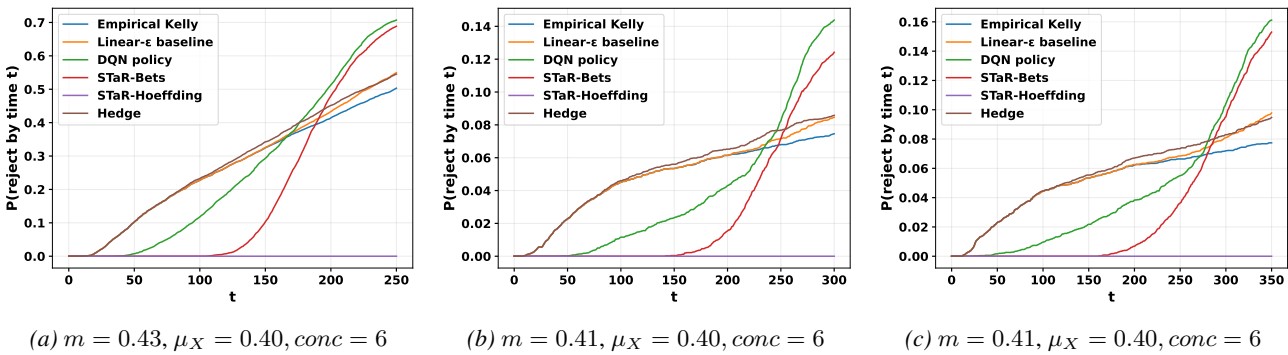

*(a) $m = 0.43, \mu_X = 0.40, conc = 6$*    *(b) $m = 0.41, \mu_X = 0.40, conc = 6$*    *(c) $m = 0.41, \mu_X = 0.40, conc = 6$*

*Figure 16.* We use the same DQN policy as in the paper and evaluate it at longer deadlines, namely $N \in 250, 300, 350$. The data-generating setting is the same Beta-mixture model used in Figure 3 of the main submission. The plots show that DQN maintains improved power under these extended horizons.

*(a) $m = 0.45, \mu_X = 0.40, conc = 6$*    *(b) $m = 0.45, \mu_X = 0.40, conc = 1$*

*(c) $m = 0.55, \mu_X = 0.60, conc = 6$*    *(d) $m = 0.55, \mu_X = 0.60, conc = 1$*

*Figure 17.* We note that DQN is flexible enough to accommodate a wide range of $\alpha$ values. Here, we set $\alpha = 0.01$. The data-generating setting is the same Beta-mixture model used in Figure 3 of the main submission. The plots show that DQN also improves power at $\alpha = 0.01$.

**Type-I error under H₀ (μ = m = 0.2)**
**N=100, world=beta, conc=1.0, α=0.05**

**Type-I error under H₀ (μ = m = 0.35)**
**N=100, world=beta, conc=1.0, α=0.05**

*(a)*

*(b)*

**Type-I error under H₀ (μ = m = 0.5)**
**N=100, world=beta, conc=1.0, α=0.05**

**Type-I error under H₀ (μ = m = 0.8)**
**N=100, world=beta, conc=1.0, α=0.05**

*(c)*

*(d)*

*Figure 18.* In all figures, the Type I error remains below $\alpha = 0.05$, as ensured by the anytime-validity of all methods. At the same time, DQN comes closer to the threshold than the competing methods, suggesting that it uses the permitted Type I error budget more efficiently, which in turn helps explain its stronger power.

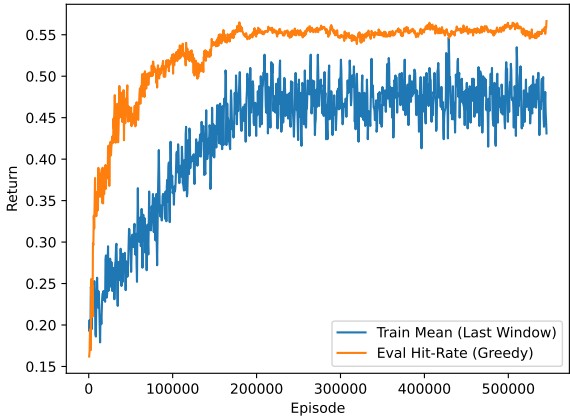

*Figure 19.* The validation hit rate in comparison with the training hit rate during DQN training.

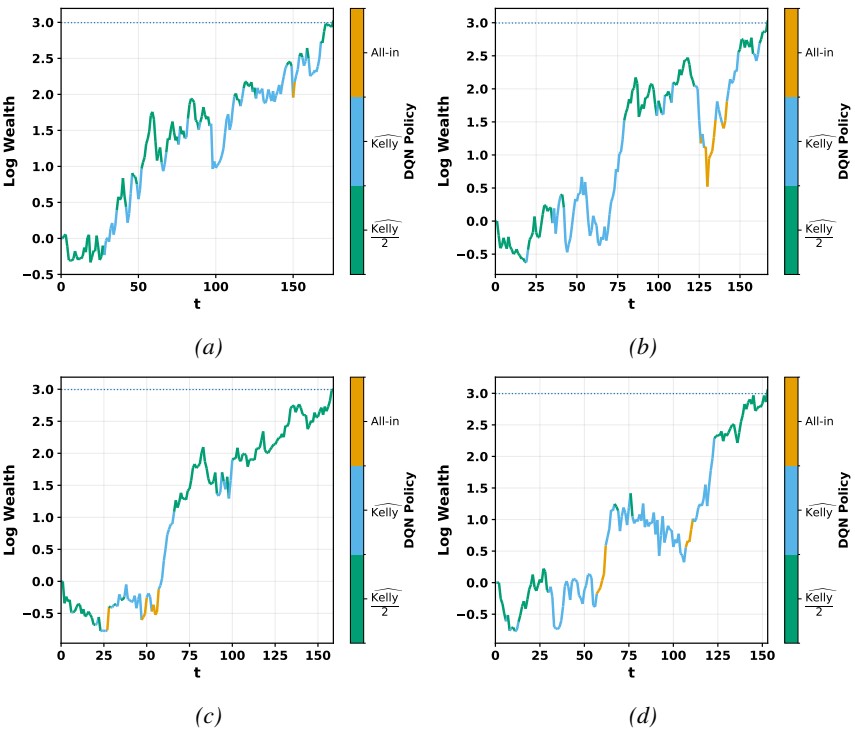

*Figure 20.* $X_i \sim \text{Beta}(\text{conc}\mu_{\text{X}}, \text{conc}(1 - \mu_{\text{X}}))$ with conc=6, $\mu_X = 0.4$, $N = 200$ with $\alpha = 0.05$, and $m = 0.41$
.

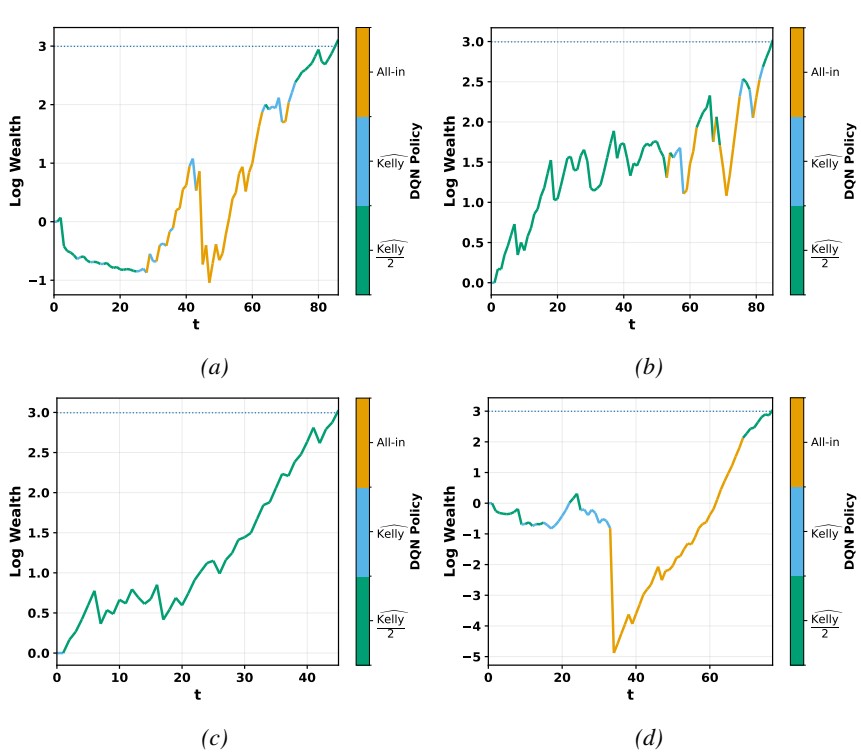

*Figure 21.* $X_i \sim \text{Beta}(\text{conc}\mu_{\text{X}}, \text{conc}(1 - \mu_{\text{X}}))$ with conc=1, $\mu_X = 0.22$, $N = 100$ with $\alpha = 0.05$, and $m = 0.28$
.

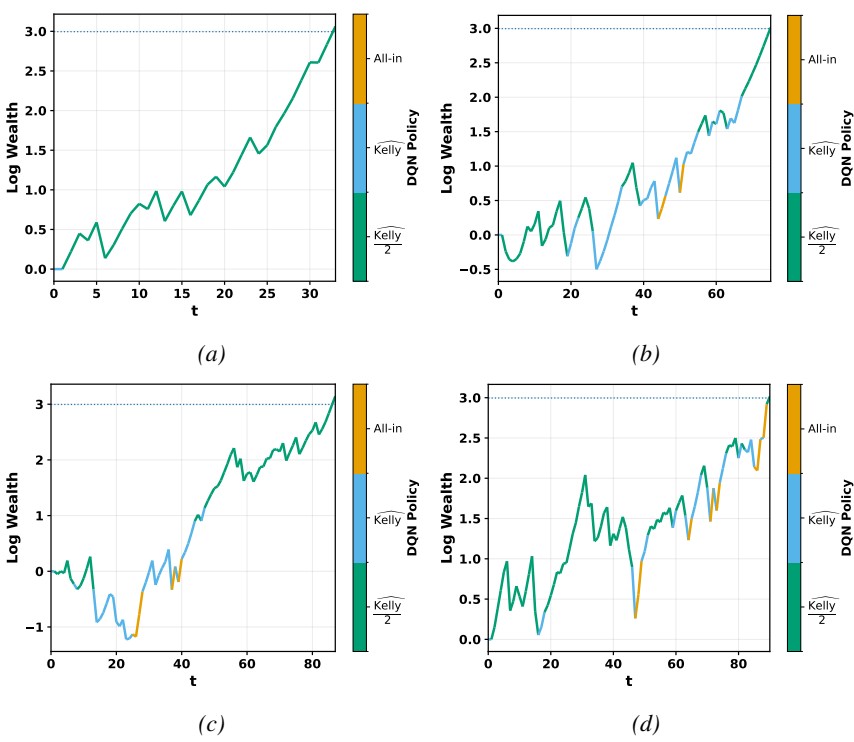

*Figure 22.* $X_i \sim \text{Beta}(\text{conc}\mu_X, \text{conc}(1 - \mu_X))$ with conc=0.5, $\mu_X = 0.71$, $N = 100$ with $\alpha = 0.05$, and $m = 0.66$
.

