# OpenReview forum: "Learning to Bet for Horizon-Aware Anytime-Valid Testing"
_ICML.cc/2026/Conference — ICML 2026 regular_

### Official Review · Reviewer_5e7w · 2026-03-10

**Soundness:** 3
**Presentation:** 2
**Significance:** 3
**Originality:** 3
**Overall Recommendation:** 4
**Confidence:** 2

**Summary:**

The paper studies anytime-valid hypothesis testing and confidence sequences under a finite horizon N. They provide a partial theoretical characterization of optimal betting in three regimes — Kelly when on schedule, aggressive when behind, defensive when ahead — and visualize this as a "phase diagram." They then train a universal DQN agent on synthetic data to learn a single betting policy across horizons, null values, and distributions, showing empirical improvements over existing baselines.

**Compliance With Llm Reviewing Policy:**

Affirmed.

**Final Justification:**

The authors have adequately addressed my main concerns. I will maintain my recommendation of weak accept.

**Key Questions For Authors:**

Check the weaknesses.

**Limitations:**

N/A.

**Strengths And Weaknesses:**

First of all, I have to confess that I am not an expert in this area,  so some of my assessments may reflect limited familiarity with the field.

Strength:
1. The problem is well-motivated and practical. Many real applications (A/B tests, clinical trials) have hard resource limits. The paper also does a good job comparing itself to prior work.
2. The phase diagram provides clean and effective intuition. The three-regime partition of the (t, log W_t) plane is easy to understand. The discussion is accessible even to readers outside this field.
3. The DQN approach is a sensible and practical solution given that the exact optimal policy is intractable.

Weaknesses:
1. All experiments are on beta or beta-mixture distributions with N ≤ 200. It will be better to evaluate the performance on other distributions (like heavy-tail distributions) outside the training distributions.
2. The computation cost of the proposed algorithm seems to be a headache. Especially, it seems that STaR-Bets and other baseline models do not require a long time of training.
3. In the numerical example, it may be better to also present results with different choices of significance level \alpha.

---

> ### Author Rebuttal · Authors · 2026-03-29
>
> We sincerely thank the reviewer for their constructive comments. In particular, we are pleased that the reviewer finds our direction well motivated and practical, our phase-diagram characterization intuitive, and the DQN approach sensible. We have added **Rebuttal Figures 3, 6, and 7**, which are accessible through **[this anonymous link](https://drive.google.com/file/d/1fA9stnCo8bHXwoERBqh5cm1iuN6Q4vl5/view?usp=sharing)**. Below, we clarify the reviewer’s concerns.
>
> ### Experimental Methodology
>
> Thank you for this helpful suggestion. We would like to clarify that the scope of this paper is intentionally the mean of bounded random variables, which already constitutes a broad and important class in the betting/CS literature. In particular, *Waudby-Smith and Ramdas (2024)* study betting-based confidence intervals and confidence sequences for bounded support. Our contribution is to study the new finite-horizon, or horizon-aware, version of this problem within that same bounded-support regime. At the same time, we do not believe that the underlying sequential-inference principles are specific to Beta/Beta-mixtures, or even to bounded data: *Wang and Ramdas (2025)* develop confidence sequences for heavy-tailed mean estimation under only a variance bound, showing that the broader CS methodology extends beyond boundedness. Thus, heavy-tail extensions of the horizon-aware control perspective are a natural next step, but they lie outside the stated scope of our paper.
>
> On evaluation, we agree that broader empirical coverage is valuable. In the rebuttal update, we additionally include logit-normal evaluations of the DQN policy, which was trained on the Beta/Beta-mixture family with randomized parameters. **Rebuttal Figure 5** shows that the DQN policy performs similarly well in this out-of-distribution setting. Furthermore, in **Rebuttal Figure 1**, we evaluate the DQN policy in the Bernoulli setting, again showing improved power.
>
> Furthermore, we also provide results with longer deadlines, specifically with $N \in \\{250, 300, 350\\}$. **Rebuttal Figure 6** demonstrates that DQN policy yields power improvements in this longer deadline setting, just as it does for shorter deadlines.
>
> ### Computational Complexity
>
> Regarding computational cost, we agree that methods such as STaR-Bets have the advantage of requiring no offline training. Our DQN incurs a one-time offline training cost of about 3 hours on a single L40S GPU, but this cost is paid only once. After training, deployment is inexpensive, since inference requires only a forward pass through a two layer MLP. Moreover, the same universal policy is reused across deadlines and distributional settings, which is precisely the advantage of our learning-based approach. Furthermore, we also have epsilon-greedy strategies that incur zero training cost while showing competitive performance.  More generally, having multiple policies is not merely an overhead in the e-process framework: distinct policies can themselves be combined via hedging, so policy diversity can be a feature rather than a drawback.
>
> ### Different Choices of Significance Level
>
> Thank you for this suggestion. In the **Rebuttal Figure 7**, we provide results for the significance level of $\alpha = 0.01$. The DQN policy continues to improve power at this significance level. We will expand the final version of our paper to include these experiments.
>
>
> We greatly appreciate the reviewer’s thoughtful comments. We hope that our responses have clarified our perspective and addressed the main concerns and we would be glad to engage further during the discussion week.
>
>
> - Ian Waudby-Smith, Aaditya Ramdas, Estimating means of bounded random variables by betting, Journal of the Royal Statistical Society Series B: Statistical Methodology, Volume 86, Issue 1, February 2024, Pages 1–27, https://doi.org/10.1093/jrsssb/qkad009
> - Wang, H., & Ramdas, A. (2025). Anytime-valid t-tests and confidence sequences for Gaussian means with unknown variance. Sequential Analysis, 44(1), 56–110. https://doi.org/10.1080/07474946.2024.2428245

---

> > ### Author Rebuttal · Reviewer_5e7w · 2026-04-03
> >
> > Thanks for the detailed review. I will maintain my positive evaluation.

---

> > > ### Author Response · Authors · 2026-04-05
> > >
> > > We sincerely thank the reviewer for their engagement during the rebuttal period and for maintaining their positive evaluation. We also note that we recently added new real-data experiments in our discussion with **Reviewer M6oR**, available **[here](https://openreview.net/forum?id=GiSCgWW0Ze&noteId=Z3tsEnbaXM)**. These additions include six new results across two scientific domains, further strengthening the empirical coverage of the paper. We appreciate the reviewer’s feedback and will incorporate the rebuttal experiments and the associated discussion into the final manuscript.

---

### Official Review · Reviewer_ZqVq · 2026-03-12

**Soundness:** 4
**Presentation:** 4
**Significance:** 3
**Originality:** 3
**Overall Recommendation:** 5
**Confidence:** 4

**Summary:**

This paper studies the problem of betting-based anytime-valid testing when the user knows there is a cap to the horizon length, and shows that knowledge of this cap can be exploited for improved power over the typical optimal Kelly betting strategy.

**Compliance With Llm Reviewing Policy:**

Affirmed.

**Final Justification:**

As highlighted in the authors' rebuttal, the generality of their framework and flexibility to incorporate other objective functions is a key strength relative to the Voracek paper, which nudges this paper up for me from a 4 to a 5.

**Key Questions For Authors:**

Wouldn’t you want the reward to account for how early you reject, not just whether or not you reject before N? This choice of binary reward shows up in the plots, where the DQN’s probability of rejection by the end is high, but the probability of rejection earlier on is lower than other methods.

**Limitations:**

Yes

**Strengths And Weaknesses:**

I find the key idea compelling and intuitive that when there is a finite horizon, one may enter states that are “ahead of schedule” or “behind schedule” in which it is beneficial to bet less or more aggressively than Kelly betting. I thought the phase diagram was a nice visualization of this, and the theory on the existence and classifications of these regimes were quite nice. I also thought the DQN was a good way to leverage this key idea methodologically, and the empirics show a substantial benefit to this approach over ignoring the finite horizon.

My only concern is about novelty, as the Voracek & Orabona (2025) recently made a similar high level observation and developed a method to leverage it. There are certainly differences between the two approaches and the theory in this paper is new, so I’m not saying there isn’t substantial novelty here, but the paper is just perhaps not as revolutionary for not being the first to make and leverage this observation about finite horizons, and the power gains of the method over the Voracek paper’s method are somewhat more modest than the gains over ignoring the finite horizon altogether.

---

> ### Author Rebuttal · Authors · 2026-03-29
>
> We thank the reviewer for their thoughtful comment and for the positive assessment of our paper. We are happy that the reviewer finds the key idea of our paper compelling and intuitive, and our methodology of using Deep-Q-Learning for this. Upon the reviewer’s requests, we have included new experiments, which are shown in **[this anonymous link](https://drive.google.com/file/d/1fA9stnCo8bHXwoERBqh5cm1iuN6Q4vl5/view?usp=sharing)**.
>
> ### Novelty
> We agree that the broad observation that a finite horizon should affect the betting policy is not unique to our paper and studied before by Voracek & Orabona (2025). Yet, our main point of departure is different: we formulate betting-based anytime-valid tests / confidence sequences with a known deadline as a finite-horizon optimal control problem.
>
> We see this control formulation as the key conceptual contribution. First, it allows us to obtain structural insight into the optimal policy, including the “ahead of schedule / on schedule / behind schedule” regimes summarized by the phase diagram. Second, it gives a principled route to improved practical procedures via reinforcement learning and also potentially approximate dynamic programming, rather than a single hand-designed horizon-aware rule. This also makes the framework flexible enough to support objectives (in the form of reward) beyond the specific one studied in the main paper.
>
> In light of this, since both our work and Voracek & Orabona (2025) exploit the finite-horizon nature of the setting, we believe a more modest gap relative to horizon-free methods is natural. Even for the same objective $\max \mathbb{P}(\tau \le N)$, our contribution is methodological: the control formulation yields structural insight into the optimal policy through the phase-diagram picture, clarifying when and why one should be ahead of schedule, on schedule, or behind schedule. At the same time, it provides a principled way to automatically discover effective state-dependent betting policies via RL (and potentially approximate dynamic programming), extending beyond hand-crafted horizon-aware algorithms such as STaR-Bets.
> Beyond this specific objective, we view the framework as more broadly useful because the objective enters through the reward. By changing the reward, one can target different inferential goals, such as earlier rejection or other risk-sensitive utilities, and again let the same control/RL methodology automatically discover the corresponding modified betting policy. Below, we elaborate on this flexibility through alternative reward designs and the different quantitative behaviors they induce.
> ### Alternative Reward Designs
> We agree that rewarding earlier rejection is a natural alternative objective, and we thank the reviewer for this suggestion. In the main paper, we use the binary reward $1{\tau \le N}$ because it exactly matches the formal objective: maximizing rejection probability by the deadline while preserving anytime validity. Thus, if the learned policy sometimes rejects later than other methods while still improving rejection probability by $N$, this reflects the chosen utility rather than a limitation of the framework. We also note that STaR-Bets shows a similar pattern of early conservativeness followed by later aggressiveness in Figure 3.b.
>
> A key advantage of the control/RL formulation is that it is not tied to this binary reward. One can instead optimize a time-weighted objective such as $w(\tau)\mathbf{1}{\tau \le N}$ with decreasing $w$, or other variance-sensitive, quantile-based, or risk-sensitive utilities.
>
> To illustrate this flexibility, we add **Rebuttal Figure 5**, which compares the original DQN with two variants trained using modified rewards. The original DQN uses the binary reward $r=1$ if rejection occurs by the deadline and $r=0$ otherwise. DQN-EB uses an early-bonus reward $r=1+(1-t/N)$, and DQN-U uses the urgency reward $r=1-t/N$, which places even more emphasis on early rejection. In the same Beta-mixture setting as Figure 3, we observe the expected tradeoff: the original DQN achieves the strongest terminal power, DQN-EB gives a middle ground, and DQN-U shifts rejection earlier at some cost in power at $t=N$. These variants may therefore be preferable when earlier rejection is more important, for example to obtain narrower confidence sequences well before the deadline. Validity is preserved in all cases because it comes from the betting construction, not from the reward used.
>
> We appreciate this suggestion because it highlights an important strength of our approach: the same framework can produce different policy behaviors under different inferential goals. We plan to expand this discussion in the final manuscript.
>
> We hope our responses have clarified our perspective and addressed the main concerns. We would be happy to engage further during the discussion week.
>
> - Voracek and Orabona, STAR-Bets: Sequential Target-Recalculating Bets for Tighter Confidence Intervals, NeurIPS 2025.

---

> > ### Author Rebuttal · Reviewer_ZqVq · 2026-04-01
> >
> > I thank the authors for their detailed response. I found the flexibility of the proposed framework helpful to have emphasized, with the demonstration that changes the utility function particularly compelling. I think my general positive opinion remains but this clarification has nudged me up to a 5.

---

> > > ### Author Response · Authors · 2026-04-05
> > >
> > > We thank the reviewer for their thoughtful feedback and engagement during the rebuttal period. We are pleased that our rebuttal helped highlight the flexibility of our framework, especially the discussion of modifying the utility function. The reviewer's comments have been very useful for us, and we will incorporate their suggestions into the final manuscript.

---

### Official Review · Reviewer_DifP · 2026-03-12

**Soundness:** 3
**Presentation:** 3
**Significance:** 2
**Originality:** 3
**Overall Recommendation:** 4
**Confidence:** 3

**Summary:**

This paper addresses the problem of designing horizon-aware, anytime-valid tests and confidence sequences for bounded means under a fixed deadline $N$. While standard anytime-valid inference allows for indefinite monitoring, it can be overly conservative when a hard resource limit exists.

The authors bridge this gap by framing horizon-aware betting as a finite-horizon optimal control problem aimed at maximizing rejection probability by the deadline while maintaining validity. The authors characterize, through a phase diagram, the structure of the optimal policy depending on the problem's difficulty.

The paper pairs theoretical results with experimental evaluations of different strategies, spanning from heuristics to online learning to deep RL.

**Compliance With Llm Reviewing Policy:**

Affirmed.

**Final Justification:**

I lean towards accepting this paper. At first I was concerned about technical novelty (which is not so strong), but after the rebuttals, the authors better emphasized the methodological role of their work (and strengthened the empirical validation). So I raised my score from 3 to 4, which better reflects my evaluation now.

**Key Questions For Authors:**

I would like the authors to highlight the technical novelty and the subsequent impact on the related literature. I'm willing to update my score if this specific concern is addressed in an adequate way.

**Strengths And Weaknesses:**

Strenghts:

- The paper is overall easy to follow, even though the reading becomes difficult in some sections
- The paper pairs theory and practice by providing theoretical tools and a comprehensive experimental
- The studied problem is natural and well-motivated: I like the fact that it combines techniques from different literatures

Weaknesses:

- While the paper tells a complete story, the technical depth and novelty are not so evident to me. I feel like most of the techniques follow standard derivations from the literature
- Experimental validation is only conducted in a simulated environment and with Beta distributions

---

> ### Author Rebuttal · Authors · 2026-03-29
>
> We sincerely thank the reviewer for their constructive feedback. We are also glad that the reviewer appreciates the comprehensiveness of our experiments, the presence of theoretical characterizations supported by empirical results, and the motivation of the problem we study. At the reviewer’s suggestion, we have included new experiments as **Rebuttal Figures 1, 3, and 5**, which are accessible at **[this anonymous link](https://drive.google.com/file/d/1fA9stnCo8bHXwoERBqh5cm1iuN6Q4vl5/view?usp=sharing)**. Their comments are very helpful in improving the final version of our paper, and we would like to clarify each concern below.
>
> ### Technical Depth and Novelty
>
> Thank you for raising this point. We emphasize that our main contributions are conceptual and methodological, and we have not attempted to achieve the “maximum generality” in terms of theory.  In particular, we formulate the design of anytime-valid tests/confidence sequences as a finite-horizon optimal-control problem over the state $(t, \log W_t)$ with the objective of maximizing rejection probability by a fixed deadline while preserving validity. This viewpoint leads to the Bellman recursion and, more importantly, to a partial structural characterization of the optimal policy. In particular, Theorem 3.1 identifies an interior regime where substantial deviations from Kelly are provably suboptimal, while Propositions 3.4 and 3.6 identify “behind-schedule” and “ahead-of-schedule” regimes where more aggressive or more defensive betting can beat Kelly. The resulting phase-diagram interpretation is the key novel insight, and it directly motivates the design of our epsilon-greedy and DQN based policies.
>
> We also see the **impact on the related literature** as **methodological**. Relative to horizon-aware heuristics such as STaR, our point of departure is an explicit control objective rather than target recalculation. That change of viewpoint is what makes  RL (and also potentially) approximate DP natural here, and it is what allows us to design learned, state-dependent, distribution-sensitive betting policies while preserving anytime-validity, since validity depends on predictability and the feasible bet constraint, not on how the policy is obtained. In that sense, the paper’s contribution is a principled optimal control lens to design new statistical methodology leveraging the well-developed literature on RL and approximate DP.  We believe this lens can be useful beyond the specific objective studied here (which is the rejection probability at timestep $N$). For example, it could be adapted to optimize CS widths at each timestep, variance-aware utilities, or other risk-sensitive criteria.
>
> In **Rebuttal Figure 5**, we illustrate this flexibility by comparing the original deadline-focused DQN with two variants that increasingly favor earlier rejection through their rewards, where we observe the expected tradeoff: the original reward achieves the highest power at the deadline, whereas the modified rewards encourage earlier rejection at some cost to terminal power. This shows that the same control/RL framework can be tuned to different inferential goals. This same perspective also explains our paper’s scope. Because the paper is primarily a methodology paper, we intentionally emphasized simple, interpretable results over maximal generality.
>
> ### Experiments on Simulated Environments and On Beta Family
>
> We also agree that broader empirical coverage would strengthen the paper. The main submission focused on Beta/Beta-mixture worlds because they provide flexible bounded families, but in the rebuttal we additionally included Bernoulli and out-of-distribution logit-normal evaluations, where the learned policy continues to improve deadline-limited power over strong baselines. Specifically, in **Rebuttal Figure 3**, we evaluate the DQN policy, which has been trained only on the Beta/Beta-mixture distributions, on logit-normal distributions. The results demonstrate that the DQN policy is robust in this challenging out-of-distribution setting. Furthermore, in **Rebuttal Figure 1**, we retrain the DQN on Bernoulli family and evaluate on Bernoulli distributions, again demonstrating the power improvements of the the DQN policy over baselines.  We will make this broader empirical scope clearer in the final revision. Also, we agree that the present evaluation is synthetic, and in the final version we will expand the discussion of this limitation and aim to add empirical validation on real data.
>
> We would like to thank the reviewer once again for the thoughtful, constructive feedback. We hope that our responses have clarified our perspective and addressed the main concerns. We would be happy to engage further during the discussion week.

---

> > ### Author Rebuttal · Reviewer_DifP · 2026-04-02
> >
> > I am very thankful to the Authors for clarifying the following: **the contribution is methodological rather than technical**. And I agree with this statement. After having re-read the submission and your rebuttal, I am now willing to raise my score to 4, as I can better see its merits. Also, I appreciated the additional and extensive experimental campaign.

---

> > > ### Author Response · Authors · 2026-04-05
> > >
> > > We are sincerely grateful to the reviewer for their thoughtful feedback and engagement during the rebuttal period. Their comments have been very useful in improving the paper, and we are glad that our clarifications and additional experiments helped make the merits of the work clearer. We will incorporate the resulting discussion into the final version.

---

### Official Review · Reviewer_M6oR · 2026-03-13

**Soundness:** 3
**Presentation:** 3
**Significance:** 3
**Originality:** 3
**Overall Recommendation:** 4
**Confidence:** 4

**Summary:**

This paper studies anytime-valid testing and confidence sequences for bounded means when the analyst faces a hard deadline N. The main idea is to view betting-based sequential inference as a finite-horizon control problem over the state (t, log W_t), where t is time and W_t is the test-martingale wealth. The paper provides a partial characterization of good horizon-aware policies: in an interior “on-schedule” region, policies that deviate substantially from Kelly betting are provably suboptimal; when the process is behind schedule, more aggressive betting can be preferable; and when it is ahead of schedule, more defensive betting can be better. Motivated by this phase-diagram view, the paper proposes both simple schedule-aware heuristics and a universal DQN policy that maps predictable summary statistics to a small discrete action set. Empirically, the learned policy improves deadline-limited rejection probability and often yields tighter horizon-aware confidence sequences than prior horizon-aware baselines such as STaR-Bets and STaR-Hoeffding, as well as simpler heuristic baselines, on synthetic Beta and Beta-mixture settings.

**Compliance With Llm Reviewing Policy:**

Affirmed.

**Key Questions For Authors:**

1. How robust are the empirical findings when the action space for λ is expanded beyond {λ/2, λ, λ_max}? Would a richer discrete set or continuous control lead to similar policies and performance improvements?

2. Can the learned DQN policy generalize beyond the Beta and Beta-mixture distributions used for training? For example, how does it perform on other bounded distributions or real-world sequential experimentation data?

3. Some theoretical results assume finite-support distributions, while experiments focus on continuous distributions. Could the authors clarify how the theory should be interpreted in relation to these experiments?

4. Could the authors include explicit empirical calibration results under the null hypothesis (e.g., type-I error or coverage plots) in the main paper to complement the power and width comparisons?

**Limitations:**

No. The paper would benefit from a clearer discussion of limitations. In particular, the empirical evaluation is entirely synthetic, the learned policy is trained and tested on closely related distribution families, and the action space used by the DQN policy is deliberately small. A discussion of how these design choices affect the applicability of the approach in real-world settings would improve the paper.

**Strengths And Weaknesses:**

The paper is technically solid overall. The betting formulation is standard and preserves anytime-validity by restricting to predictable bets within a safe range and by using mixtures when hedging across schedules. The authors also avoid overclaiming optimality and instead present partial structural results that motivate practical policies. The appendix provides substantial implementation details and experimental settings, which supports reproducibility.

However, there remains a gap between the theoretical analysis and the empirical evaluation. Some theoretical results assume finite-support distributions, whereas the experiments focus on continuous Beta and Beta-mixture distributions. While this does not invalidate the experiments, the connection between theory and empirical setup could be clearer. In addition, the learned policy operates over a very small discrete action set {λ/2, λ, λ_max}, meaning the phase diagram and empirical gains are tied to this coarse action space rather than a richer policy class. Finally, I would have appreciated more explicit empirical reporting of null calibration or coverage properties, even though validity holds by construction.

---

> ### Author Rebuttal · Authors · 2026-03-29
>
> We thank the reviewer for their positive assessment of our work and constructive feedback. We are glad that the reviewer found our paper technically solid, and we appreciate their suggestions regarding action-space richness, out-of-distribution evaluation and the further clarifications regarding the theory/experiments connection. In response, we added four new experiments in **Rebuttal Figures 1, 2, 3, and 4**, accessible at **[this anonymous link](https://drive.google.com/file/d/1fA9stnCo8bHXwoERBqh5cm1iuN6Q4vl5/view?usp=sharing)**. Below we clarify each point.
>
> ### Action Space
> Our choice of the  3-action set was due to the fact that it is the smallest discrete set that directly matches the three theory-motivated regimes in the phase diagram: conservative, approximately Kelly, and aggressive betting. That said, we agree that richer action sets may yield larger power gains. To investigate this question, we trained an expanded 9-action version and compared it with our original 3-action policy in **Rebuttal Figure 2**. Empirically, the larger action space achieved power improvements similar to those of the 3-action policy relative to existing baselines, but it behaved somewhat differently: the 9-action version attained a higher rejection probability at earlier times, but a slightly lower rejection probability at the deadline.
>
> Our interpretation is that, under our current synthetic training setup, the benefit of expanding the action space may be limited if the total training time (~3hours) and synthetic dataset size are held constant. Intuitively, in this sparse-reward RL setting, a larger action space makes optimization harder and increases both data and training requirements. Thus, with fixed compute, data and a simple DQN architecture, the larger-action policy may simply be less well optimized. We view this as preliminary evidence and we will include a more thorough exploration in the final version including architectural variations, larger and more diverse training, as well as continuous RL experiments.
>
> ### Generalization Beyond Beta Family
> Thank you for raising this point. We trained over randomized Beta and Beta-mixture worlds because they provide a broad family of bounded continuous distributions with varying mean, concentration, and mixture structure. In **Rebuttal Figure 3**, we also evaluate the Beta-trained policy on out-of-distribution logit-normal data and observe that the performance gains persist, with the DQN policy improving its power over existing baselines. This suggests that policies learned on Beta distributions can transfer to some other continuous test distributions. In **Rebuttal Figure 1**, we also show that the same framework performs well on Bernoulli families when trained on them, suggesting that the control/RL formulation is not tied to a single parametric family.
>
> In the final version, we plan to broaden training and evaluation to include both continuous and discrete synthetic distributions, as well as their mixtures, and we will aim to add empirical validation on real data.
>
>
> ###  The Finite-Support Assumption in the Theory
> Our theoretical goal was intentionally partial: to establish structural facts showing that, depending on the state $(t, \log W_t),$ it can be suboptimal to always follow Kelly. There is also an important distinction between the results. Theorem 3.1 does not require finite support. The finite-support restriction is used only in the sufficient conditions of Propositions 3.4 and 3.6, where it enables clean type-based large-deviation arguments and an explicit $O(\log T/T)$ finite-sample slack term. Our goal there was not to claim a complete optimality theorem for all continuous experiments, but to give a partial structural characterization of when deviating from Kelly can help. We expect the same qualitative picture to extend beyond finite alphabets: one route is to replace finite-alphabet Sanov arguments by the general version, which would yield a similar conclusion with an unspecified $o(1)$ slack term. Another possible route is a discretization+approximation argument to attempt to recover explicit slack terms following the approach in *Baldasso et al. (2023)*. Additionally, to better bridge theory and experiments, we now also include Bernoulli experiments in **Rebuttal Figure 1**, where the finite-support assumptions apply, and the similar power gains appear.
>
> ###  Explicit Null Calibration
> Thank you for the suggestion. We have added explicit null-calibration plots showing $P(\text{reject by } t)$ under $H_0$ for several null values in the **Rebuttal Figure 4**. As expected from the theoretical validity guarantees, all methods remain below $\alpha = 0.05$. The DQN tends to come closer to the $\alpha$-budget than more conservative baselines while remaining valid, which helps explain its improved power under the alternative.
>
> - Baldasso, R., Oliveira, R.I., Pereira, A. et al. A Proof of Sanov’s Theorem via Discretizations. J Theor Probab 36, 646–660 (2023).

---

> > ### Author Rebuttal · Reviewer_M6oR · 2026-04-03
> >
> > Thank you for the detailed rebuttal. I appreciate that the authors directly addressed several of my questions by adding experiments on a larger action space, additional synthetic distribution families, and explicit null-calibration analyses. These additions improve the paper. However, my core concern is not only whether the method generalizes across closely related synthetic settings, but whether the observed gains matter beyond them. Because this is ultimately a validation-oriented empirical paper rather than a primarily theoretical contribution, its significance depends heavily on whether the phenomenon carries over to more realistic settings.

---

> > > ### Author Response · Authors · 2026-04-05
> > >
> > > We thank the reviewer for their engagement and for this suggestion. While our paper also includes theoretical analysis, we agree that evidence of gains beyond the synthetic families would further strengthen the empirical significance of our methodology. To address this directly, we conducted new experiments on two real-data domains comprising six unseen data sources: genome-wide DNA methylation beta values and daily relative humidity measurements. The new results are accessible through **[this anonymous link](https://drive.google.com/file/d/13NWgMXBHg46FWuTuqRePGZZ8bW5kX2Y3/view?usp=sharing)**.
> > >
> > > **Evaluation setup.** In each trial, we draw $N=100$ observations with replacement from the empirical distribution of a real dataset, run the sequential test $H_0:\mu=m$ at level $\alpha=0.05$, and repeat this for $5{,}000$ trials to obtain rejection curves $\mathbb{P}(\text{reject by time } t)$. Importantly, the DQN policy was trained exclusively on synthetic Beta and Beta-mixture distributions and was not fine-tuned on any real data. This setup preserves the i.i.d. evaluation setting of our method while testing transfer to real distributions from different scientific domains.
> > >
> > > **Data domain 1: DNA methylation** ([NCBI GEO: GSE33896](https://www.ncbi.nlm.nih.gov/geo/query/acc.cgi?acc=GSE33896), **Rebuttal Figures 8-10**). We use genome-wide CpG methylation beta values. Each observation is a continuous beta value in [0, 1], where 0 indicates a fully unmethylated CpG site and 1 indicates a fully methylated site. We test three datasets spanning adipose-derived stem cells, induced osteocytes, and a rhabdomyosarcoma cell line. These empirical distributions are strongly heterogeneous, with pronounced skewness and bimodality, and arise from a completely different data-generating process than the synthetic families used in training. We provide histograms for the empirical distributions along with the power results.
> > >
> > >
> > > **Data domain 2: Daily relative humidity** ([NOAA U.S. Climate Reference Network](https://www.ncei.noaa.gov/access/crn/products.html), **Rebuttal Figures 11-13**). We use daily average relative humidity values from three stations spanning distinct climates: desert (Arizona, Yuma), mountain (Colorado, Boulder), and temperate Northeast (New York, Millbrook). We scale the raw measurements from [0%, 100%] to [0, 1] by dividing by 100. These yield right-skewed, near-symmetric, and mildly left-skewed distributions, respectively. Because our framework assumes i.i.d. observations, we evaluate these data by resampling individual daily measurements with replacement from the observed record. This isolates transfer to the real marginal distribution rather than robustness to temporal dependence, which is outside the current scope of this work.
> > >
> > >
> > > Across these six real-data settings, the DQN policy achieves the highest rejection probability by the deadline in 5/6 cases and is very close to the best method in the remaining case, while maintaining valid null calibration in all cases. In other words, a policy trained only on synthetic Beta/Beta-mixture families transfers without retraining to real data from two distinct application domains. This provides concrete initial evidence that the observed gains extend beyond synthetic settings and reflect a more general horizon-aware betting strategy.
> > >
> > > In the final version of our paper, we will further strengthen the empirical evaluation of our method by adding more extensive real-data evaluations, broadening the synthetic training families, and augmenting our synthetic training corpora with real data. At the same time, we believe these preliminary real-data results provide meaningful evidence that the benefits of our method extend beyond synthetic settings.
> > >
> > > We again thank the reviewer for this suggestion. We were glad to carry out these additional experiments and will incorporate the resulting discussion into the final version of the paper.

---

### Decision · Program_Chairs · 2026-04-30

**Decision:**

Accept (regular)

**Comment:**

Though the technical analysis itself is not particularly novel, I believe the reviewers generally agree that the main contribution of this work is methodological rather than technical. The use of a reinforcement learning framework, in particular DQN, to address this finite-horizon betting problem is both interesting and well-motivated. The empirical results demonstrate the practical performance of this framework, and notably, the learned policy successfully recovers the predicted optimal action structure observed in the theoretical analysis. Taken together, these aspects constitute a solid contribution to the machine learning community.